# Aerosol Scattering Effects on Water Vapor Retrievals over the Los Angeles Basin

Zhao-Cheng Zeng[1,2], Qiong Zhang[1], Vijay Natraj[3], Jack S. Margolis[4], Run-Lie Shia[1], Sally Newman[1], Dejian Fu[3], Thomas J. Pongetti[3], Kam W. Wong[1,3], Stanley P. Sander[3], Paul O. Wennberg[1] and Yuk L. Yung[1]

[1]Division of Geological and Planetary Sciences, California Institute of Technology, Pasadena, CA 91125, USA
[2]Institute of Space and Earth Information Science, The Chinese University of Hong Kong, Hong Kong, China
[3]Jet Propulsion Laboratory, California Institute of Technology, Pasadena, CA 91109, USA
[4]1842 Rose Villa St., Pasadena, CA 91107, USA

*Correspondence to*: Zhao-Cheng Zeng (zcz@gps.caltech.edu)

**Abstract.** In this study, we propose a novel approach to describe the scattering effects of atmospheric aerosols in a complex urban environment using water vapor ($H_2O$) slant column measurements in the near infrared. This approach is demonstrated using measurements from the California Laboratory for Atmospheric Remote Sensing Fourier Transform Spectrometer on the top of Mt. Wilson, California, and a two-stream-exact single scattering radiative (2S-ESS) transfer (RT) model. From the spectral measurements, we retrieve $H_2O$ slant column density (SCD) using 15 different absorption bands between 4000 and 8000 cm$^{-1}$. Due to the wavelength dependence of aerosol scattering, large variations in $H_2O$ SCD retrievals are observed as a function of wavelength. Moreover, the variations are found to be correlated with aerosol optical depths (AOD) measured at the AERONET-Caltech station. Simulation results from the RT model reproduce this correlation and show that the aerosol scattering effect is the primary contributor to the variations in the wavelength dependence of the $H_2O$ SCD retrievals. A significant linear correlation is also found between variations in $H_2O$ SCD retrievals from different bands and corresponding AOD data; this correlation is associated with the asymmetry parameter, which is a first-order measure of the aerosol scattering phase function. The evidence from both measurements and simulations suggests that wavelength-dependent aerosol scattering effects can be derived using $H_2O$ retrievals from multiple bands. This understanding of aerosol scattering effects on $H_2O$ retrievals suggests a promising way to quantify the effect of aerosol scattering on greenhouse gas retrievals and could potentially contribute towards reducing biases in greenhouse gas retrievals from space.

## 1 Introduction

Greenhouse gas (GHG) observations from space provide unprecedented global measurements of column GHG concentration, facilitating inference of carbon fluxes on regional scales (Yoshida et al., 2011; Crisp et al., 2012). However, atmospheric aerosol scattering has been shown to have a considerable impact on the retrieval of GHGs from space-based observations in the near infrared (Aben et al., 2007; Yoshida et al., 2011; O'Dell et al., 2012). A study by O'Dell et al. (2012) showed that

the error budget in the satellite retrievals of column-averaged dry-air mole fraction of $CO_2$ ($XCO_2$) is dominated by systematic errors due to imperfect characterization of cloud and aerosol properties. Previous studies also showed that the bias can be greatly mitigated by incorporating simple aerosol properties into the retrieval state variables (Butz et al., 2009; Guerlet et al., 2013). It is therefore crucial to characterize the aerosol properties for the GHG retrieval algorithm. However,

aerosols have complex types and size distributions and are highly variable in number density. Their optical properties are very difficult to measure directly (Seinfeld and Pandis, 2006). The global ground-based aerosol monitoring network, AERONET (Holben et al., 1998), has been providing high-accuracy measurements of total aerosol optical depth (AOD) from the ultraviolet to the near infrared, but is sparsely distributed, suggesting that it would be useful to have further means of constraining aerosol optical properties.

Water vapor ($H_2O$) has absorption features across the electromagnetic spectrum, from the ultraviolet to the infrared. In a non-scattering atmosphere, $H_2O$ retrievals from different absorption bands would give exactly the same value. However, due to wavelength-dependent aerosol scattering (Eck et al., 1999; Zhang et al., 2015), the different light path changes in different absorption bands result in discrepancies in $H_2O$ retrievals. This variation in $H_2O$ retrievals from different bands can therefore provide information on aerosol properties. Based on this principle, we propose a novel approach to describe aerosol

scattering effects using the variation in $H_2O$ retrievals from multiple spectral bands. This approach is illustrated using data from the California Laboratory for Atmospheric Remote Sensing (CLARS), which continuously collects high-resolution spectra in the near infrared. A two-stream-exact single scattering (2S-ESS) radiative transfer (RT) model is used to simulate the observations and explain the physical mechanism behind the proposed approach. The additional information on aerosol optical properties gained by applying this approach could be used to improve retrievals of GHGs from space in the presence

of aerosols.

In Section 2, a detailed description of the CLARS-FTS is presented. Section 3 introduces the 15 different bands chosen for retrieving $H_2O$ from CLARS measurements. In Section 4, the correlation between variations in $H_2O$ retrievals from different bands and corresponding AOD data from AERONET-Caltech is discussed. In Section 5, the role of aerosol scattering on the variations in $H_2O$ retrievals is illustrated using the 2S-ESS RT model. Discussion and conclusions are presented in Sections

6 and 7, respectively.

## 2 CLARS

The CLARS-FTS is located near the top of Mt. Wilson at an altitude of 1670 m a.s.l. overlooking 28 land target sites in Los Angeles (Supplemental material Table 1 and Figure 1 of Fu et al., 2014; Wong et al., 2015; Wong et al., 2016). It offers continuous high-resolution spectral measurements from 4000 to 8000 cm$^{-1}$. As shown in Figure 1, CLARS-FTS has two

operating modes: the Spectralon Viewing Observation (SVO) mode and the Los Angeles Basin Surveys (LABS) mode. In SVO mode, the FTS looks at a Spectralon$^{®}$ plate adjacent to the FTS to observe the reflected solar spectrum from the free

troposphere above Mt. Wilson, which is usually above the planetary boundary layer (PBL) and relatively immune to aerosol scattering. In LABS, the FTS looks down at target sites in the LA basin to observe the reflected sunlight, which travels through a long light path within the urban PBL (Figure 1) and undergoes absorption and scattering by gas molecules and aerosols. This observation geometry by LABS mode makes CLARS-FTS measurements not only highly sensitive to the atmospheric composition in the PBL but also very susceptible to influence by aerosol scattering and absorption. The CLARS measurement technique from Mt. Wilson mimics geostationary satellite observations of reflected sunlight, which are governed by a sun to surface and surface to instrument optical path geometry. The geostationary observations can be used to retrieve GHG mixing ratios (e.g., Xi et al., 2015).

Slant column density (SCD), the total number of absorbing molecules per unit area along the optical path, is retrieved for $H_2O$ using the version 1.0 operational retrieval algorithm of CLARS-FTS (Fu et al., 2014), developed based on the Gas Fitting tool (GFIT) algorithm that has been widely used for the Total Carbon Column Observing Network (TCCON) network (Toon et al., 1992; Wunch et al., 2011). Surface reflection is included in the latest version of GFIT but aerosol scattering is not taken into account. Thus, errors in the SCD retrieval are largely due to light path changes caused by aerosol scattering, and can be used as a proxy for aerosol loading. A detailed description of CLARS-FTS, including the observation system, the measurement sequence, the operational retrieval algorithms and the operational data products can be found in Fu et al. (2014). The 15 different bands used in this study for retrieving $H_2O$ SCDs from CLARS-FTS are introduced in Section 3.

## 3 Spectral bands for $H_2O$ retrievals

$H_2O$ has absorption features across the electromagnetic spectrum including many absorption lines in the near infrared as well as significant continuum absorption. Within the spectral range of CLARS-FTS observations, we carefully select 15 spectral intervals containing dominant $H_2O$ absorption lines, as shown in Figure 2 and Table 1, across the spectral range from 4000 to 8000 $cm^{-1}$ for retrieving $H_2O$ SCD. Each interval is 5 to 14 $cm^{-1}$ wide and contains two to nine moderately strong absorption lines. Rodgers (2000) shows that information analysis is a powerful tool which can be used in the channel selection process. Therefore, we conduct information analysis by calculating the information content (IC) for each band selected for retrieving $H_2O$ SCD. IC is calculated as $-\ln|\mathbf{I} - \mathbf{A}|/2$, where I is the identity matrix and $\mathbf{A}$ is the averaging kernel, a measure of the sensitivity of the retrieval state to true state variables. A detailed theoretical description of the information analysis can be found in Su et al. (2015). The RT model used here is the numerically efficient 2S-ESS RT model (Spurr and Natraj, 2011). The detailed settings for the 2S-ESS model are described in Section 5. In the RT model, absorptions by the dominant gas molecules in the atmosphere, including $H_2O$, $CO_2$, $N_2O$, $CO$, $CH_4$, $O_2$, $N_2$ and HDO, are considered by using an *a priori* atmospheric profile obtained from the National Center for Environmental Prediction (NCEP)-National Center for Atmospheric Research (NCAR) reanalysis data (Kalnay et al., 1996). AOD observations from AERONET, and SSA and g

from MERRA-GOCART are included in the RT model (details are provided in Section 5). Here, we assume that only one state variable, i.e., $H_2O$ SCD, is retrieved. Therefore, the IC values shown in Figure 2 are for $H_2O$ retrieval only; they are indicators of the precision of $H_2O$ retrieval in the selected bands. For any band with IC value equal to $x$, as many as $e^x$ different atmospheric $H_2O$ states can be distinguished (Rodgers, 2000). The purpose of the IC calculation is to show the

sensitivity of each absorption band to the variation of $H_2O$ SCD. While fitting the CLARS-FTS measured spectra, we retrieve more state variables including other trace gas abundances ($CO_2$, $CH_4$, $CO$, and $N_2O$) and continuum shape parameters (Fu et al., 2014). From the IC results shown in Figure 2, we can see that the average IC is high (about 6 on average) and similar among all bands, which indicates very high retrieval precision compared with the *a priori* information (Rogers, 2000). $H_2O$ SCDs retrieved from these 15 bands will be shown in the following section.

**4 $H_2O$ SCD retrievals from CLARS and its correlation with AOD**

**4.1 Daily variation**

Figure 3 shows examples of daily CLARS $H_2O$ SCD retrievals on March 01, 2013 and September 28, 2013. On these two days, there were dense observations for the West Pasadena target retrieved using the 15 bands shown in Figure 2. Observations between 9:00 and 15:00 local time are shown, when the solar zenith angles (SZAs) are less than 60°, similar to

the majority of satellite observations. This time period also includes the local overpass times at around 10:00 for SCIAMCHY (Bovensmann et al., 1999) and around 13:30 for GOSAT, OCO-2 and TanSat (Liu et al., 2011). In particular, we compare retrievals from SVO (Figure 3(a)), which is above the PBL (Newman et al., 2013) and therefore relatively unaffected by aerosol scattering, with those from the West Pasadena target (Figure 3(b)), a location in the Los Angeles basin that is influenced by aerosol scattering. The $H_2O$ SCD retrievals from SVO are nearly identical across different wavelengths,

and the small differences (about 14.2% and 8.2% of those from West Pasadena for March 01 and September 28, respectively, in terms of standard deviation of $H_2O$ SCD retrievals) may be attributed to the band-to-band inconsistency of line parameters. However, $H_2O$ SCD retrievals for West Pasadena show significantly larger variation (about 7 times and 12 times those from SVO for March 01 and September 28, respectively, in terms of standard deviation of $H_2O$ SCD retrievals) across different wavelengths. These retrieval differences are much larger than can be attributed to spectroscopic uncertainties alone, and

reflect the wavelength dependence of aerosol scattering in the boundary layer. To quantify the variation in the $H_2O$ SCD retrievals from the 15 bands, the standard deviation ($\sigma$) of the retrievals is calculated by,

$$\sigma = \sqrt{\sum_{i=1}^{n}[(s_i - \bar{s})^2]/(n-1)}, \tag{1}$$

where n=15 is the number of bands, $s_i$ is the SCD retrieval for band i, and $\bar{s} = (\sum_{i=1}^{n} s_i)/n$ is the mean. The standard deviations in Figure 3 (c) show that the variations in $H_2O$ SCD retrievals monotonically increase throughout the day. As

shown in Figure 3 (d), the AOD data on these two days from AERONET-Caltech shows a typical pattern for the LA basin. AOD increases from the morning to the afternoon. Note here that the AERONET station measures mainly in the visible and

near-infrared wavelengths from 340 to 1020 nm. However, we assume that these measurements are also good proxies for AOD at longer wavelengths in the near infrared. The increasing trend of AOD during the course of the day corresponds well to that of the standard deviations of $H_2O$ SCD retrievals. This correlation from daily measurements shows the potential of constraining the AOD using the standard deviation of $H_2O$ SCD retrievals. However, apart from aerosol scattering, the daily variation of standard deviations of $H_2O$ SCD retrievals can also be influenced by differences in observation geometry, such as SZA and relative azimuth angle (AZA), and variations in planetary boundary layer height (PBLH). All of these parameters are changing during the day. These effects are included in the discussion below.

### 4.2 Seasonal variation

To quantitatively compare the variations in $H_2O$ SCD retrievals and AOD, we choose the daily mean of the data between 12:00 and 14:00 local time when there is generally less haze or fog. This time is also coincident with the local crossing time of the two currently operating GHG observation satellites, GOSAT (Yoshida et al., 2011) and OCO-2 (Crisp et al., 2012). We focus on a two-hour time period to limit the effect of other parameters on the $H_2O$ retrieval standard deviation, such as solar geometry and PBLH. The closest (temporal and spectral) AERONET AOD data are used as the concurrent AOD data for the CLARS retrievals. If the measurement time difference is more than 30 min, then the CLARS data are not used. AERONET-Caltech has AOD measurements in seven wavelengths from 340 to 1020 nm, and we choose the data at 1020 nm, which is the closest to the wavelengths used for retrieving $H_2O$ in this study (from 1289 to 2196nm). Furthermore, aerosol optical path (AOP) values are calculated by multiplying the vertical path AOD data by air mass factors, which are derived from the SZA and viewing zenith angle of CLARS at West Pasadena (83.1°). Following this procedure, 68 daily mean data pairs are available in 2013 after excluding days (1) that are cloudy according to the images from the visible camera looking at the target and (2) in which there are fewer than three valid observations. We separate the data into two different time periods, the winter-spring season (December to May) and the summer-autumn season (June to November) by considering the different dominant wind directions between winter and summer in Pasadena (Conil and Hall, 2006; Newman et al. 2016). In the summer, the prevailing winds come from the southwest across the basin; in the winter, the winds come from the northeast across the mountains and deserts. Different wind patterns (Newman et al., 2016) suggest that the dominant aerosol types during these two time periods may be different. For each of these two time periods, we expect the CLARS observation geometry and PBLH at noon time to be similar across the days, and therefore assume their effects on $H_2O$ retrievals to be similar. In fact, it will be clear from the RT model simulation in Section 5 that the contributions to the variation in $H_2O$ retrievals from other factors, such as solar geometry and PBLH, are actually much smaller than those from aerosol scattering, even though the geometry and PBLH change a lot during a day.

Figure 4 shows significant linear correlations between the AOP value and the standard deviation of $H_2O$ SCDs for the two time periods, both with $R^2$ around 0.5. However, the slopes from linear regression between the AOP and the standard

deviation of $H_2O$ SCD retrievals are different. In summer-autumn, the regression slope, an indicator of the degree of light path change due to aerosol scattering relative to the change in aerosol loading, is about one half of that in winter-spring, indicating that for the same AOD, the variation in $H_2O$ SCD retrievals in summer-autumn is about twice that in winter-spring. $H_2O$ abundance shows significant seasonal variation in the LA basin, and previous studies have shown, for example, that

aerosol optical properties change dramatically with relative humidity (e.g. Thompson et al., 2012). In fact, results from control experiments using the RT model, shown in Section 5, show that the difference in $H_2O$ abundance and aerosol phase function, indicated by the asymmetry parameter, are responsible for the difference in slope between the winter-spring and summer-autumn periods.

## 5 Simulations using 2S-ESS RT model

In this section, we use the 2S-ESS RT model to simulate the measurements and quantify the role of aerosol scattering in the variation in $H_2O$ SCDs retrieved from CLARS. The 2S-ESS model performs an exact computation of the single scattering using all moments of the phase function, while the multiply-scattered radiation is calculated using the two-stream approximation. This model has been used for GHG remote sensing in several previous studies (Xi et al., 2015; Zhang et al., 2015; Zhang et al., 2016).

We simulate the spectral radiance observed by CLARS-FTS for the West Pasadena target. The settings for this model are largely the same as those used by Zhang et al. (2015). Some essential settings and modifications are as follows. In this RT model, (1) the *a priori* atmospheric profile is obtained from NCEP-NCAR reanalysis data (Kalnay et al., 1996). The profile has 70 layers from the surface up to 70 km; (2) absorption coefficients for all absorbing gas molecules are derived from the HITRAN version 2008 database (Rothman et al., 2009); (3) the optical depth for each layer is calculated using the Reference

Forward Model (Dudhia et al., 2002); (4) the surface reflection is assumed to be Lambertian with a surface albedo of 0.23, as measured by Fu et al. (2014) for West Pasadena; (5) Rayleigh scattering by air molecules is considered in this RT model; (6) the observation geometry, including the viewing zenith angle for the West Pasadena target, the daily SZA, and AZA on March 01, 2013 are included in the model; (7) the aerosol scattering phase function in the model is assumed to follow the Henyey-Greenstein type phase function (Henyey and Greenstein, 1941). Climatological aerosol compositions, as percentages

of total optical depth, for five types of aerosol (black carbon, organic carbon, sulfate, dust, and sea salt) are obtained from the Modern Era Retrospective analysis for Research and Applications (MERRA) aerosol reanalysis database (Rienecker et al., 2011; Buchard et al., 2015). A more detailed description of the data can be found in Connor et al. (2016). The aerosol single scattering properties, including single scattering albedo (SSA) and asymmetry parameter (g), were computed using the GOCART model (Colarco et al., 2010, Chin et al., 2002). These properties were tabulated for black carbon, organic carbon,

sulfate, dust, and sea salt aerosol types, with hygroscopic effects included where appropriate. The average values of the scattering parameters are shown in Table 2. Using the compositions of the five composite MERRA aerosols and their

scattering properties, the effective SSA and g of aerosol scattering are calculated, respectively, as the composition-weighted sum, and then incorporated into the 2S-ESS RT model. Figure 5 shows the monthly-averaged climatological aerosol compositions for the five composite MERRA aerosols. We can see that, in general, sea salt dominates in the summer-autumn period while dust dominates in the winter-spring period; (8) unlike Zhang et al. (2015), in this study, the average hourly PBLH data measured over late spring in 2010 in LA (Newman et al., 2013) are used. Unlike $CO_2$, the $H_2O$ mixing ratio varies dramatically with altitude in the atmospheric column (Seinfeld and Pandis, 2006). In the LA basin, a large portion of $H_2O$ is concentrated within the PBL. Therefore, the PBLH is an important parameter in modeling the effects of scattering on the $H_2O$ retrieval. The model output radiance is convolved using the CLARS-FTS instrument line shape with full width at half maximum (FWHM) = 0.022cm$^{-1}$ (Fu et al., 2014). The spectral resolution is set to be 0.06 cm$^{-1}$, and the corresponding instrument maximum optical path difference is 5.0 cm. The signal-to-noise ratio is assumed to be constant at 300. We perturb the simulated spectra with Gaussian white noise.

The wavelength range covered by AERONET-Caltech measurements is from 340 to 1020 nm; however, the wavelengths of the 15 $H_2O$ absorption bands used in this study, ranging from about 1280 to 2200 nm, are outside the AERONET wavelength range. For the sake of calculations in the 2S-ESS RT model, the AOD data in these 15 bands were extrapolated using the Ångström exponent law (Seinfeld and Pandis, 2006; Zhang et al., 2015). This law is given by:

$$\frac{\tau}{\tau_0} = (\frac{\lambda}{\lambda_0})^{-k} \tag{2}$$

where $\lambda$ and $\tau$ are the wavelength and the corresponding AOD to be interpolated, respectively; and $\lambda_0$ and $\tau_0$ are the reference wavelength and the corresponding AOD from AERONET, respectively; and $k$ is the Ångström exponent. $k$ is obtained by applying linear regression using the logarithmic form of Equation (2), on the AERONET AOD measurements in the six different bands (340 nm, 380 nm, 440 nm, 500 nm, 870 nm and 1020 nm). Examples of applying this law to the AERONET AOD measurements are shown in Supplemental Material Figure 1, from which we can see that the wavelength dependence of the total AOD, a combination of different types of aerosols, generally follows the above exponent law. The extrapolated AOD data on March 01, 2013 for the 15 bands are included in the RT model, assuming non-zero AOD is evenly distributed vertically and horizontally in the PBL.

## 5.1 Simulations of Daily Variation

To quantify the influence of aerosol scattering on the $H_2O$ SCD retrievals, we simulate the bias observed by CLARS-FTS by (1) using the 2S-ESS RT model to generate synthetic spectral radiance data for the 15 chosen bands, and (2) fitting the synthetic spectral data and retrieving $H_2O$ SCD based on Bayesian inversion theory (Rodgers, 2000) using the forward 2S-ESS RT model with the same configuration, but with AOD set to zero and held constant, as in Zhang et al. (2015). This approach approximately simulates the influence of neglecting aerosol scattering on the retrieved $H_2O$ SCDs by CLARS. The

fitting process employs the Levenberg-Marquardt algorithm (Rodgers, 2000). The state vector element to be retrieved from the inversion approach is the scaling factor, which is the ratio of retrieved $H_2O$ SCD to the assumed "truth" data obtained from NCEP reanalysis. Supplementary Material Figure 2 is a schematic diagram of the retrieval algorithm which is based on the 2S-ESS RT model and Bayesian inversion theory.

We perform three experiments to demonstrate the effect of aerosol scattering on the variations in $H_2O$ SCD retrievals (Figure 6). In the first experiment, denoted as Case I (Figure 6(a)), the AOD data vary during the day in the same way as the AERONET-Caltech measurements, while in the second (control) experiment, denoted as Case II (Figure 6(b)), the AOD data are fixed at the clear-day level, in which the lowest AOD across the year is used for all hours across the day. In the third (also control) experiment, denoted as Case III (Figure 6(c)), the AOD is set to be zero for all hours across the day. In these

three cases, the changes of AOD in the forward modeling are made only when we generate the synthetic spectral radiance (see also the Supplementary Material Figure 2). For inverse modeling (when we use the RT model to produce the Jacobian matrix), we keep the AOD constant at zero to approximately simulate the influence of neglecting aerosol scattering on the retrieved $H_2O$ SCDs by CLARS.

The results for simulated $H_2O$ SCD retrieval scaling factors are shown in Figures 6(a), (b) and (c), respectively, for these

three experiments. The scaling factors (f) are mean-centered by subtracting the mean to clearly show the variations in the data (scaling factors before mean-centering are shown in Supplementary Material Figure 3). The mean-centered scaling factor is calculated as: $\hat{f}_i = f_i - \bar{f}$ for i = 1 to 15, where $f_i$ is the scaling factor for band i, $\hat{f}_i$ is its mean-centered scaling factor, and $\bar{f}$ is the mean of the scaling factors. From Figure 6(a), we can see that the variation in the simulated $H_2O$ SCD retrievals increases with increasing AOD from the morning to the afternoon, similar to what we see from the CLARS observations in

Figures 3(b) and (c). The scattering effect is stronger in the afternoon than in the morning, as shown in Figure 6(b), even though the AOD is the same for all hours; this effect is probably due to the changes in SZA and AZA from the morning to the afternoon. However, since the results from the control experiments show much smaller diurnal changes, seen in Figures 6(b) and (c), than the first experiment, aerosol scattering must be the dominant cause of the variations in $H_2O$ SCD retrievals.

Further confirmation of this is provided by Figure 6(d), which shows the comparison between CLARS measurements and the

three RT model experiments in terms of normalized standard deviations of $H_2O$ SCDs (data before normalization are shown in Supplementary Material Figure 4). As with the CLARS measurements, the standard deviations of the scaling factors are calculated using equation (1). To emphasize the relative change in variations of $H_2O$ SCDs between measurements and simulations, their standard deviations (σ) are normalized to lie between 0 and 1. The normalized standard deviations are calculated as $\tilde{\sigma}_t = (\sigma_t - \sigma_{min})/(\sigma_{max} - \sigma_{min})$, where $\sigma_t$ is the standard deviation, $\tilde{\sigma}_t$ is the normalized standard deviation at

time t, and $\sigma_{max}$ and $\sigma_{min}$ are the maximum and minimum standard deviations, respectively, throughout the day. This normalization is independently implemented for measurements and simulations. When normalizing the CLARS

measurements, the half-hourly means of the data are calculated to obtain the maximum and minimum. From Figure 6(d), we can see that the increasing trend from measurements is very similar to that from the simulations from the first experiment, while the trend is much smaller for the control experiment. The major difference between the results from the different cases can be attributed to the effect of aerosol scattering in the PBL. Hence, we conclude that aerosol scattering is the dominant

factor contributing to the variations in $H_2O$ SCD retrievals, and the correlation between AOD and standard deviations of $H_2O$ SCD retrievals from measurements is robust. Therefore, the $H_2O$ SCD retrievals from CLARS can potentially be used for constraining the aerosol properties in the LA basin.

### 5.2 Simulations of Seasonal Variation

To investigate the sensitivity of the correlation between variations in $H_2O$ SCDs and the corresponding AOP (as shown in

Figure 4) to aerosol scattering properties, we reproduce the correlation using the 2S-ESS RT model with input AERONET AOD data, daily aerosol compositions (at 13:00 local time; monthly means shown in Figure 5) derived from MERRA aerosol reanalysis data, and aerosol scattering properties (shown in Table 2), including SSA and g, computed by the GOCART model for the 68 days (discussed in Section 4.2) at 13:00 local time. The retrieval algorithm is the same as that described in Section 5.1. After obtaining the retrieved scaling factor for $H_2O$ SCD for each of the 68 days, the simulated $H_2O$

SCD retrieval is calculated as the product of the scaling factor and the $H_2O$ abundance truth, which is set to be the mean of the $H_2O$ SCD retrievals from the 15 bands used by the CLARS-FTS operational retrieval algorithm.

The result is shown in Figure 7(a). We can see that the strong linear correlations between the standard deviations of $H_2O$ SCDs and the corresponding AOP are well reproduced with higher $R^2$ for both winter-spring and summer-autumn periods. However, the slopes from the linear regression in the simulations are slightly overestimated (by about 20%) relative to those

for the CLARS data (as shown in Figure 4), probably due to the uncertainties of the input climatology aerosol compositions or single scattering properties. Interestingly, the ratio of the slope from the winter-spring to that from the summer-autumn periods from the 2S-ESS model, which is about 2.0, agrees well with that from CLARS observations. On the other hand, the difference between the two slopes becomes very small in Figure 7(b). This suggests that the difference in $H_2O$ abundance between the winter-spring and summer-autumn periods is responsible for the slope difference.

Similar control experiments implemented to examine the impact of variations of SSA and g are shown in Figures 7(c) and (d), respectively. We can see that the results are similar to those in Figure 7(a), suggesting that the variations of SSA and g are not the key contributors to the variations of the standard deviations of $H_2O$ SCDs. However, the difference of slopes in Figure 7(d) are slightly smaller compared to those in Figure 7(a), indicating that g might contribute to the small difference which still exists in Figure 7(b) even when the AOD does not vary. To further investigate the impact of g on the slope,

Figure 7(e) shows the result from the 2S-ESS model with the same settings as those in Figure 7(a) but with g reduced by 50%; we can see that the slopes for the two time periods become smaller (by about 13%), indicating that the scattering

effects become stronger when g is reduced. This is because, when g is smaller, the aerosol scattering phase function becomes less forward peaked and more light is scattered at scattering angles around 40 degrees, which is the scattering angle for CLARS at noon time. Small aerosols, such as organic carbon, black carbon and sulfate from urban pollution and biomass burning have much smaller SSA and g, as shown in Table 2, suggesting that the slope will also be small during events that are dominated by these small aerosols. From these results, we can conclude that the strong linear correlation between AOD and variations in $H_2O$ SCDs is robust, and that g is the main contributor to the ratio of the changes (the slope in linear regression) between them.

## 5.3 Retrieval of AOD using the $H_2O$ absorption bands

The purposes of this section are to (1) demonstrate that accurate AODs can be retrieved from the spectra data using the 15 $H_2O$ absorption bands, and (2) investigate the impact of AOD retrieval on the process of retrieving $H_2O$ SCDs in the $H_2O$ absorption bands. Given the fact that the operational retrieval algorithm for CLARS-FTS developed based on the GFIT algorithm does not take aerosol scattering into account, we make the above calculations by conducting a realistic numerical simulation study using the 2S-ESS RT model. The advantages of this numerical simulation study are that (1) the truth state vector is known and we can directly assess the accuracy of the retrievals, and (2) control experiments can be conducted by minimizing the influences from other factors except the one that is investigated. The calculations are implemented using noon data for the 68 days shown in Figure 4.

 (1) Simultaneous retrieval of $H_2O$ SCDs and AODs

In the retrieval algorithm illustrated in Supplementary Material Figure 2, we first produce synthetic spectra using the 2S-ESS RT model with input AOD from AERONET and SSA and g from MERRA-GOCART. We then set $H_2O$ SCD scale factor and AOD as the state variables, and simultaneously retrieve them using the 15 $H_2O$ absorption bands. The ICs for the 15 $H_2O$ absorption bands are shown in Table 1. The retrieval results are shown in Figure 8(a) and 8(b). From the comparison between retrieved and true AODs (Figure 8(a)), we see that the retrieved AOD agrees well with the truth in the $H_2O$ absorption bands for all months ($R^2$=0.93; RMSE=0.0051). From the simultaneously retrieved $H_2O$ SCD scale factor and AOD averaged over the 15 bands (Figure 8(b)), we see that their retrieval errors show a similar pattern; when the $H_2O$ SCD scale factor diverges from unity, the difference between the retrieved and true AOD also increases. When $H_2O$ SCD becomes smaller, the observed radiance increases since fewer photons are absorbed in the $H_2O$ bands. On the other hand, when AOD becomes smaller, the observed radiance decreases, since the aerosol scattering effect becomes weaker and fewer photons will be scattered to the observer in the $H_2O$ bands. Therefore, we can see that the AOD and $H_2O$ SCD retrievals show the same pattern, whereby smaller AOD retrievals coincide with smaller $H_2O$ SCD retrievals, so that their effects on the observed radiance largely cancel out.

(2) Retrieval of $H_2O$ SCDs when AODs are perfectly known

We set the $H_2O$ SCD to be the only state variable and assume that the AOD is perfectly known. Comparison of the retrieved $H_2O$ SCD scale factors from three different cases are shown in Figure 8(c). In Case A, aerosol scattering is not considered; in Case B, $H_2O$ and AOD are simultaneously retrieved; in Case C, the AOD is perfectly known. We can see that (1) $H_2O$ SCD can be accurately retrieved if we have perfect knowledge of AOD, and (2) when we retrieve AOD and $H_2O$ SCD simultaneously, the variations in retrieved $H_2O$ SCD scale factors are largely reduced compared to the case when aerosol scattering is not considered.

Based on the above results from a realistic numerical simulation study, we conclude that (1) accurate AODs and $H_2O$ SCDs can be simultaneously retrieved from the spectral data in the 15 $H_2O$ absorption bands, and (2) variations in retrieved $H_2O$ SCD scale factors are largely reduced when we retrieve AOD simultaneously compared to that when aerosol scattering is not considered.

## 6 Discussion

### 6.1 Assumptions

When comparing the $H_2O$ SCD retrievals from CLARS for the West Pasadena target with AOD data from AERONET-Caltech in Figures 3 and 4, two assumptions are made: (1) aerosol distribution is homogeneous between West Pasadena and Caltech. Since the distance between them is 5 km, which is small in terms of the whole LA basin, we expect the AOD variations at Caltech and West Pasadena to be similar. However, local topography is complex and the AOD horizontal distribution can be inhomogeneous (Lu et al., 1994). The differences between these two locations may contribute to the scatter of data and slightly lower $R^2$ in Figure 4(b) compared to Figure 4(a); (2) The changes in CLARS observation geometry and PBLH at noon across the days in 2013 within either winter-spring or summer-autumn time periods are small, as well as their effects on the variations in $H_2O$ SCD retrievals from multiple bands. This assumption is supported by the results from RT modeling that contributions from factors other than AOD are very small even though the diurnal variations in observation geometry and PBLH are large.

### 6.2 Limitations

There are some limitations of the approach outlined here. First, in the 2S-ESS RT model, we use the climatological aerosol types and size distributions in LA because of the lack of temporally resolved measurements of aerosol properties in this region. In the future, more cases with different compositions of aerosol should be incorporated into the RT model to further explore this proposed approach. Second, due to the limited number of concurrent observations of $H_2O$ SCD and AOD, the correlation is only explored for two different time periods in 2013, the winter-spring seasons and summer-autumn seasons.

Correlation between the two data sets over smaller time intervals can provide more detailed information on the aerosol properties. Finally, only CLARS data for the West Pasadena target are explored since this location is closest to the AERONET-Caltech station, which is regarded as the ground truth. Unfortunately, the Caltech campus is out of sight for the CLARS observatory near the top of Mt. Wilson. More data from other CLARS targets may be explored in the future.

**7 Conclusions**

We illustrate the robust ability of using multi-wavelength retrievals of water vapor slant columns to describe aerosol scattering effects. We apply this approach to $H_2O$ SCD retrievals from 15 different absorption bands using spectral data observed by CLARS-FTS. We explore the correlation between the variation in $H_2O$ SCDs and concurrent AOD measurements from AERONET-Caltech, and use the 2S-ESS RT model to quantitatively demonstrate the dominant role of

aerosol scattering on the variation in $H_2O$ SCD data and justify the potential of using $H_2O$ retrievals to quantify aerosol scattering effects. We find that: (1) the wavelength dependence of aerosol scattering can be clearly observed by comparing the CLARS $H_2O$ SCD retrievals between SVO and LABS modes; (2) a significant linear correlation is found between the standard deviations of $H_2O$ SCDs and AOD data. Results from RT modeling are consistent with the observations, demonstrating that aerosol scattering is the dominant cause of the variation in $H_2O$ SCDs. The ratio of AOD changes to

variations in the retrieved $H_2O$ SCD depends strongly on the asymmetry parameter of the aerosol phase function. These two pieces of evidence justify our proposed approach to derive the aerosol scattering effects using $H_2O$ retrievals, providing a promising way to quantify the effect of aerosol scattering on GHG retrievals and potentially contribute towards reducing biases in GHG retrievals from space.

**Acknowledgments**

We thank M. Gunson and A. Eldering for stimulating discussions and support, and M. Gerstell for proofreading the manuscript. Part of the research in this study was performed at the Jet Propulsion Laboratory (JPL), California Institute of Technology (Caltech), under a contract with the National Aeronautics and Space Administration (NASA). Support from the Caltech KISS Megacity project, the NIST GHG and Climate Science Program and NASA's Carbon Cycle Science Program through the JPL is gratefully acknowledged. Z.-C. Zeng was supported by a postgraduate studentship for overseas academic

exchange from the Chinese University of Hong Kong. We thank Jochen Stutz and his staff for their effort in establishing and maintaining the AERONET Caltech site. The AERONET data for this paper can be downloaded online (http://aeronet.gsfc.nasa.gov); CLARS-FTS data are available from the authors upon request.

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

**Table 1**. Central wavelengths and band widths of the 15 $H_2O$ absorption bands and the corresponding information contents for retrievals of only $H_2O$ SCD (IC_a) and for simultaneous retrievals of $H_2O$ SCD and AOD (IC_b).

| Central wavelength (cm-1) | Band width (cm-1) | IC_a | IC_b |
|---|---|---|---|
| 4554.0 | 8.0 | 6.34 | 8.61 |
| 4568.0 | 12.0 | 6.44 | 8.47 |
| 4596.5 | 9.0 | 6.36 | 8.27 |
| 4632.0 | 8.0 | 5.89 | 6.72 |
| 4645.0 | 10.0 | 6.41 | 8.04 |
| 4703.0 | 14.0 | 6.44 | 9.55 |
| 5859.0 | 8.0 | 6.13 | 8.11 |
| 5910.0 | 6.0 | 5.60 | 6.29 |
| 6505.5 | 5.0 | 5.20 | 5.36 |
| 6534.0 | 12.0 | 6.15 | 7.58 |
| 6550.0 | 10.0 | 6.40 | 8.73 |
| 6618.0 | 10.0 | 6.52 | 9.40 |
| 7712.5 | 7.0 | 5.97 | 7.61 |
| 7738.0 | 12.0 | 6.20 | 8.23 |
| 7760.0 | 10.0 | 6.13 | 7.72 |

**Table 2**. Aerosol single scattering properties, computed using the GOCART model[a], for the five types of aerosols (black carbon, organic carbon, sulfate, dust and sea salt) used in the study. Single Scattering Albedo (SSA) and asymmetry parameter (g) are averaged values over the 15 $H_2O$ absorption bands.

| | black carbon | organic carbon | sulfate | dust | sea salt |
|---|---|---|---|---|---|
| SSA | 0.03 | 0.77 | 0.97 | 0.94 | 0.99 |
| g | 0.10 | 0.25 | 0.35 | 0.71 | 0.80 |

[a]Colarco et al. (2010) and Chin et al. (2002)

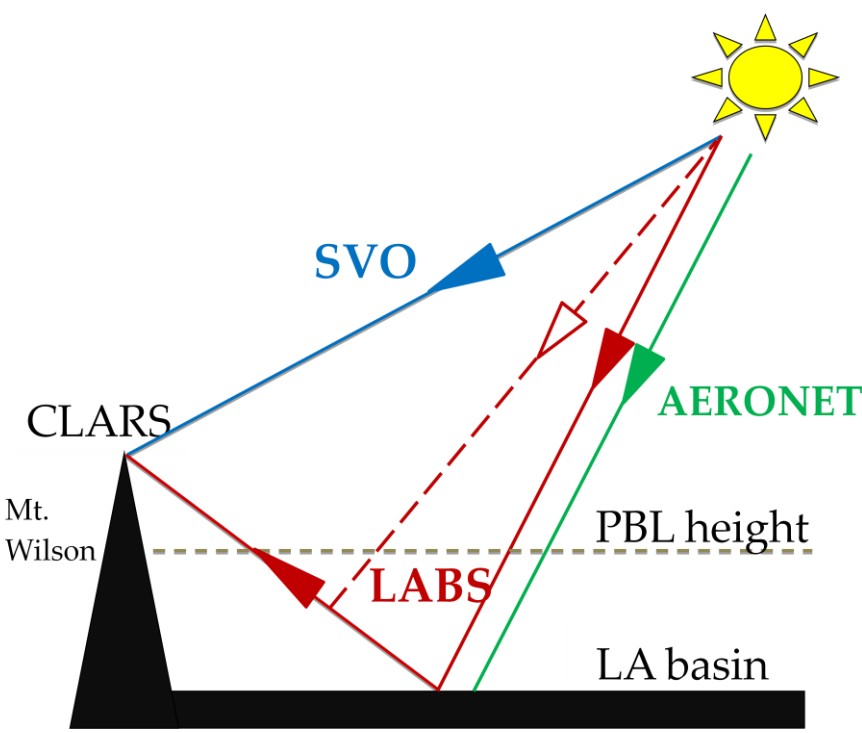

**Figure 1.** Schematic diagram of CLARS-FTS measurement geometries for West Pasadena and the AERONET site at Caltech. CLARS-FTS has two modes of operation, including Los Angeles Basin Survey mode (LABS; in solid red) and the Spectralon Viewing Observation mode (SVO; in blue). An example of light path change due to aerosol scattering along the path from the basin to the mountain top is illustrated (in dotted red). Also shown is the light path of AERONET-Caltech (in green).

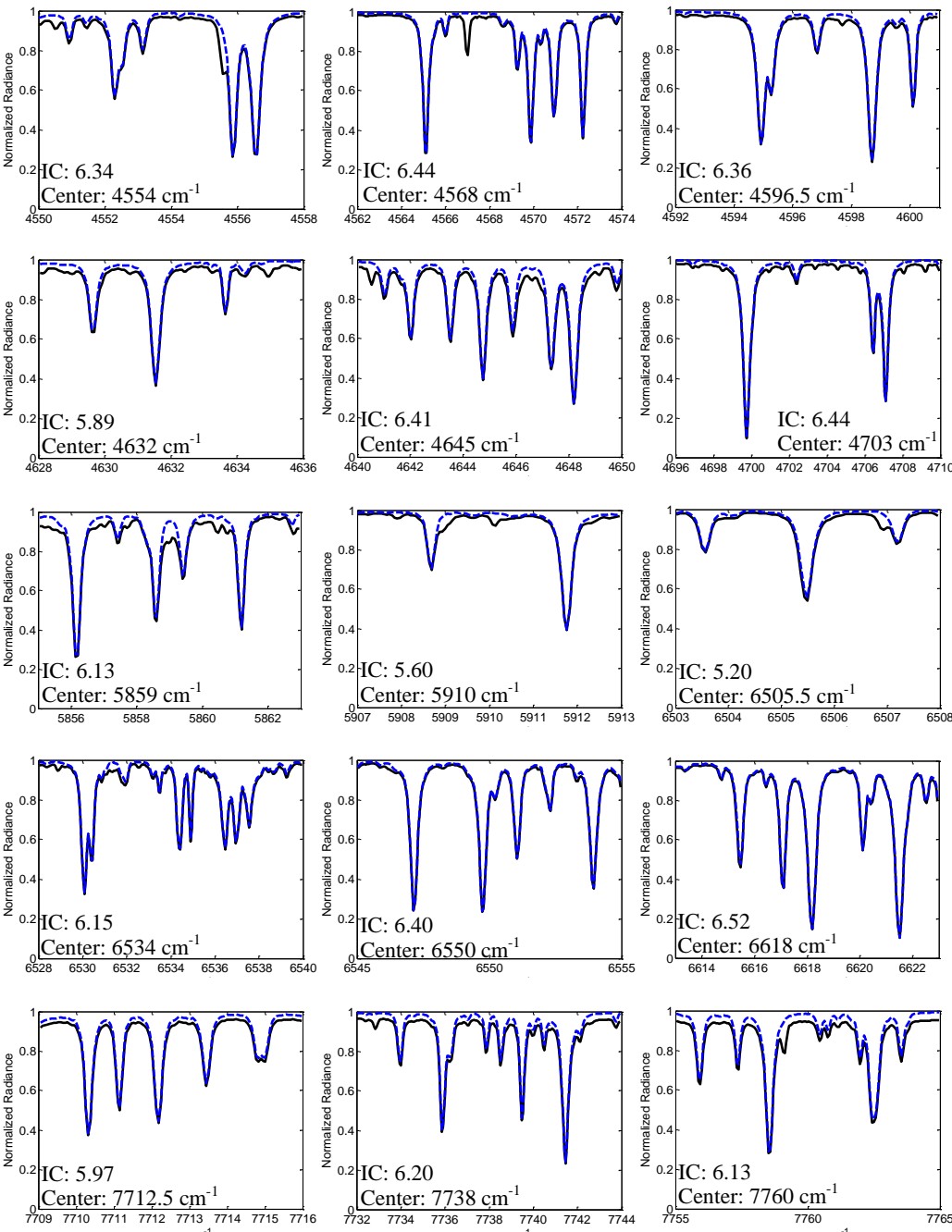

**Figure 2.** The normalized radiances, obtained by dividing the spectra by the maximum radiance, of the 15 $H_2O$ absorption bands selected for retrieving $H_2O$ SCDs from CLARS-FTS measurements. These radiances are spectral fits using the CLARS-FTS measurements in West Pasadena on March 01, 2013 with a solar zenith angle (SZA) of 41.45 °. Solid black curves are fits to the spectra, including contributions of all trace gases and solar lines, from spectral measurements by the FTS and dashed blue curves are the estimated contribution of $H_2O$ absorption to the spectral fits. Contributions of other species in these spectral regions are not shown. Central wavelength and information content (IC) value of each band used for retrieving $H_2O$ content are also indicated.

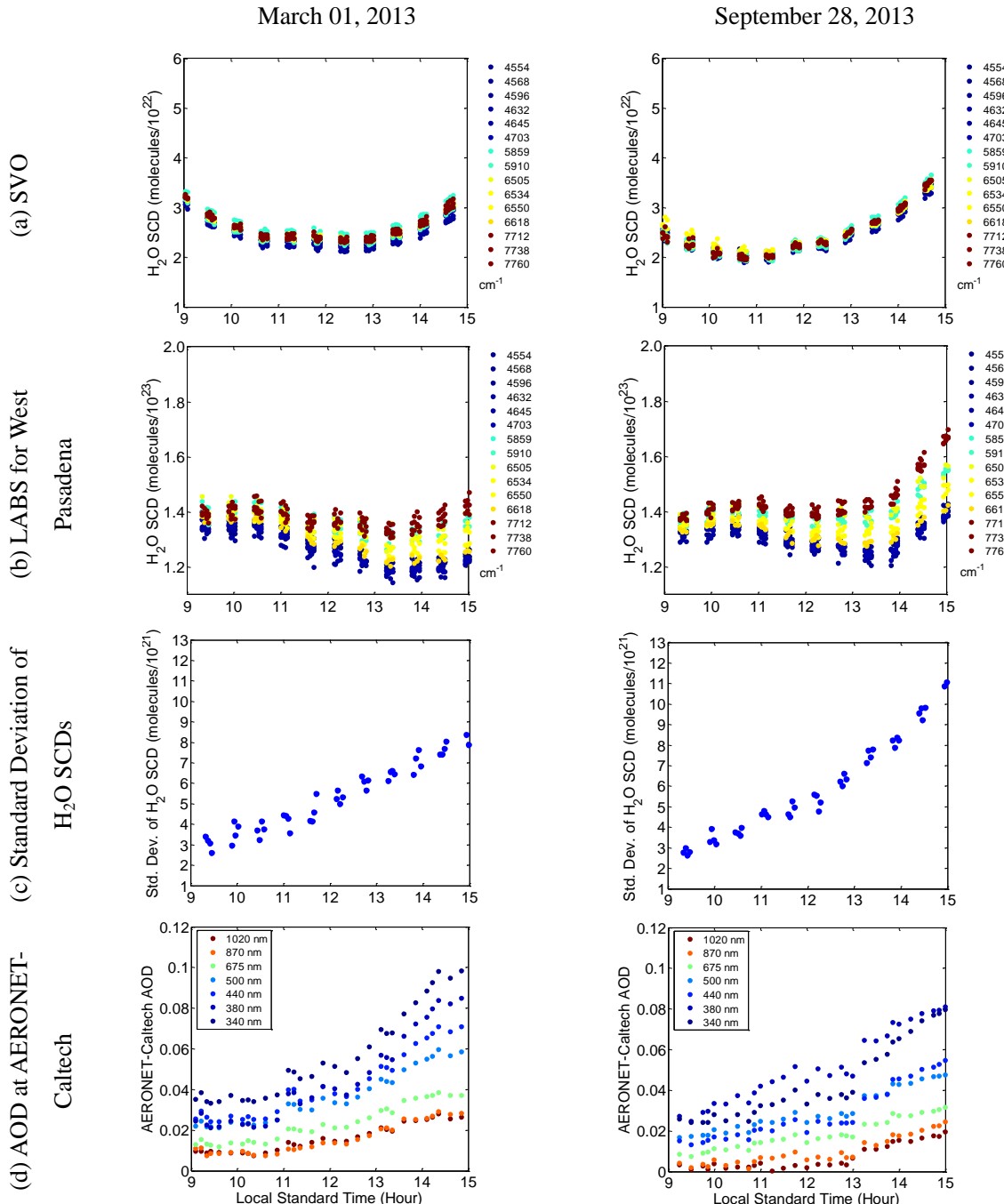

**Figure 3.** Daily Variations of CLARS $H_2O$ SCD retrievals for the West Pasadena target and AOD measurements from AERONET-Caltech station on March 01, 2013 (left column) and September 28, 2013 (right column). $H_2O$ SCD retrievals from SVO mode are shown in panel (a), and from LABS mode for West Pasadena are shown in panel (b). The corresponding standard deviations of $H_2O$ SCD retrievals, a measure of the degree of variation in the retrievals, are shown in panel (c) and 5 the AOD measurements from AERONET-Caltech are shown in panel (d).

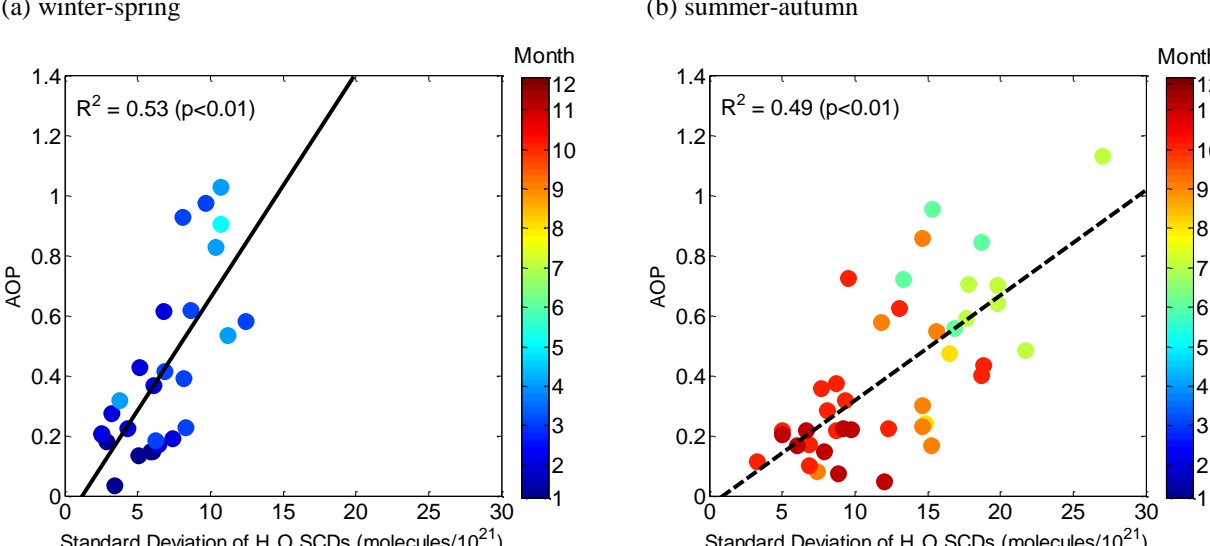

**Figure 4**. Correlation between daily averaged standard deviation of $H_2O$ SCDs, a measure of retrieval differences, from 12:00 to 14:00 and the corresponding averaged AOP, calculated by scaling AOD data (1020 nm) from AERONET based on CLARS geometry, for two time periods in 2013. (a) Winter and spring, including January to May, in which the coefficient of determination ($R^2$) is 0.53 and [slope, intercept] = [0.08±0.03, -0.09±0.21] with 95% confidence bounds from linear regression. No December data from AERONET are available in 2013. (b) Summer and autumn from June to November, in which $R^2$ is 0.49 and [slope, intercept] = [0.04±0.01, -0.03±0.16]. In total, there are 68 days of daily mean data available in 2013, out of which 27 days are for winter-spring and 40 days are for summer-autumn.

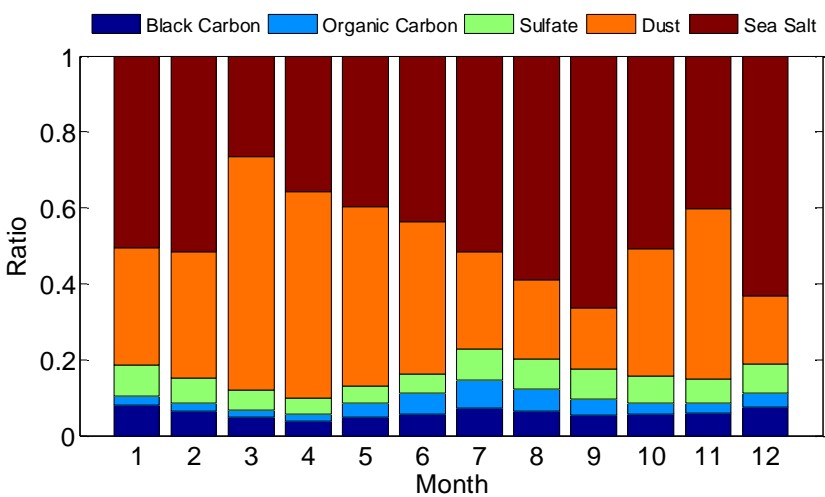

**Figure 5.** Monthly-averaged climatological aerosol composition (as percentages of total optical depth averaged over the 15 $H_2O$ absorption bands) for the five composite MERRA aerosols (black carbon, organic carbon, sulfate, dust and sea salt) in the day time (local times 13:00).

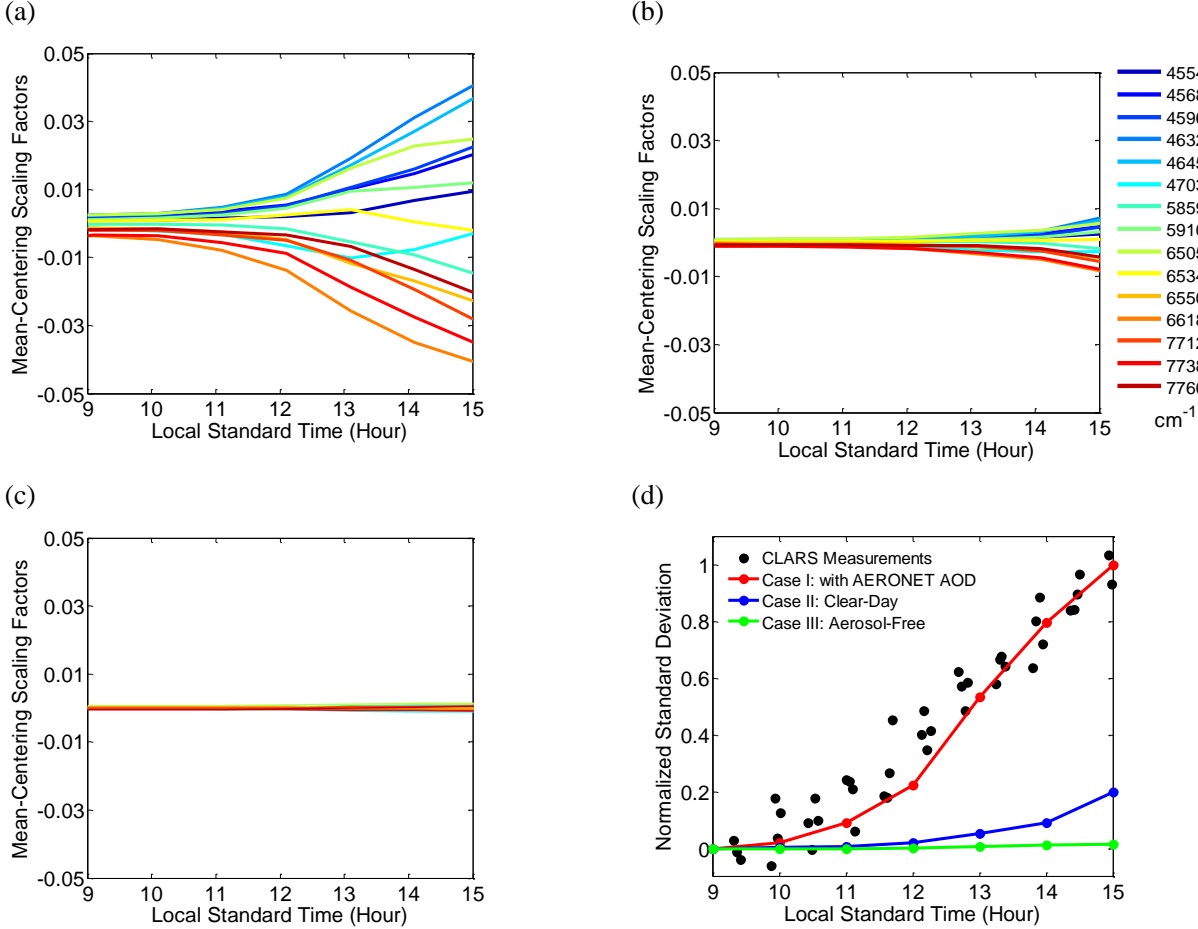

**Figure 6.** (a) Case I: scaling factors for $H_2O$ SCDs retrieved from the simulated synthetic spectral radiance of the 15 chosen bands using the 2S-ESS RT model with AOD data from AERONET-Caltech on March 01, 2013. The scaling factors are mean-centered by subtracting the mean to clearly show the variations in the retrievals; (b) Case II: same as (a) except that the AOD is fixed at the clear-day level, in which the lowest AOD in 2013 is used for all hours across the day; (c) Case III: same as (a) except that the AOD is set to be zero for all hours across the day; (d) Comparison between CLARS measurements and results from the three RT model experiments in (a), (b) and (c) in terms of standard deviations, a measure of variations in $H_2O$ SCDs retrieved from the 15 chosen bands. The standard deviations are normalized to be between 0 and 1 for both measurements and simulations. The half-hourly mean of the CLARS data is calculated to obtain the maximum and minimum for the normalization.

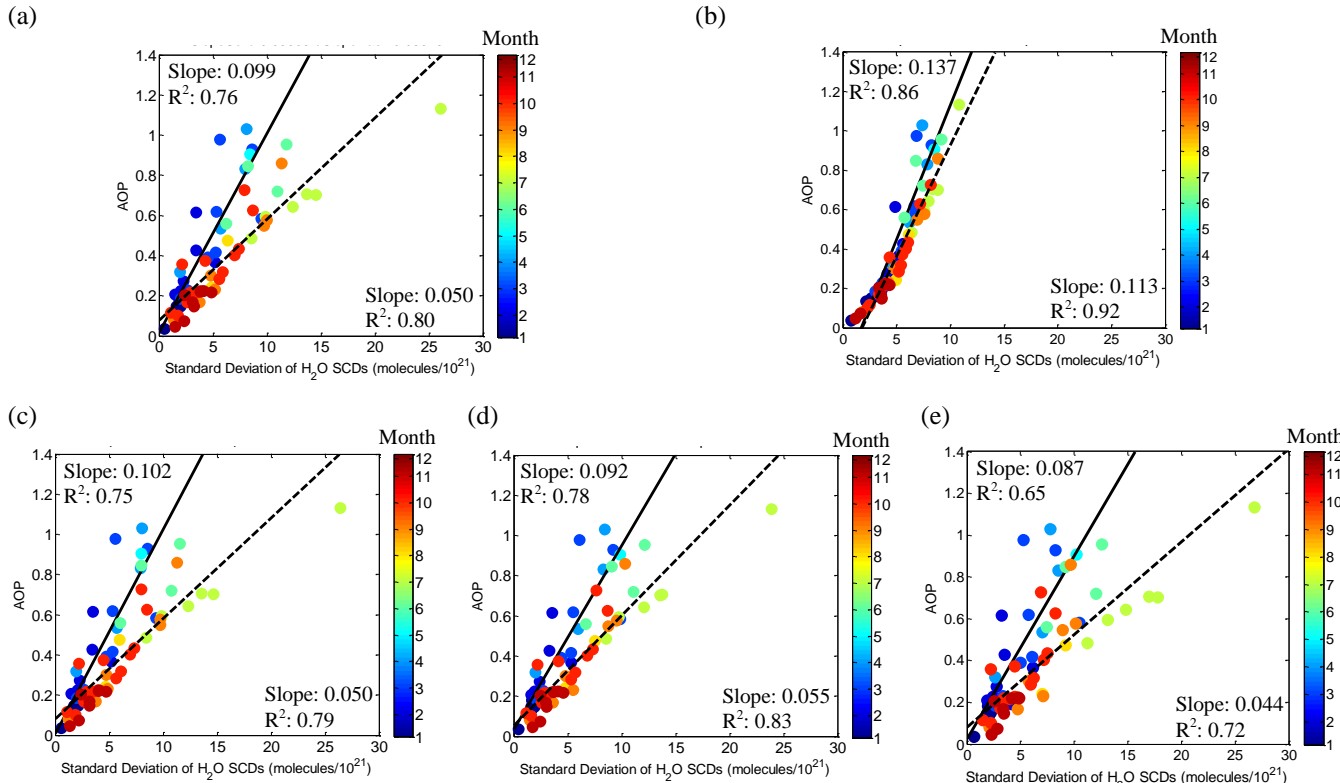

**Figure 7**. Reproduction of the correlation, as shown in Figure 4, between daily averaged standard deviation of $H_2O$ SCDs at noon and the corresponding averaged AOP for two time periods in 2013, using the 2S-ESS RT model. The input AOD data are obtained from AERONET, aerosol compositions are derived from MERRA aerosol reanalysis data, and the related scattering properties, including single scattering albedo (SSA) and asymmetry parameter (g), are computed using the GOCART model. The slopes and $R^2$ from the linear regression for winter-spring period (solid line) and summer-autumn period (dash line) are shown on the upper-left and bottom-right corners, respectively. Results from five experiments are presented, including (a) the linear correlation reproduced by the 2S-ESS model with averaged AOD data between 12:00 and 14:00 and compositions for the five composite aerosols (black carbon, organic carbon, sulfate, dust, and sea salt) at 13:00 for the 68 days; (b) same as (a) except that the $H_2O$ SCD truth is fixed for all days at the mean value for the 68 days in 2013, in order to examine the impact from variability in $H_2O$ abundance; (c) same as (a) except that the SSA is fixed for all days at the mean value for the 68 days in 2013, in order to examine the impact from SSA variations; (d) same as (a) except that g is fixed for all days at the mean value for the 68 days in 2013, in order to examine the impact from g variations; (e) same as (a) except that g for each of the 68 days is set at half of the original value, in order to examine its contribution to the linear correlation.

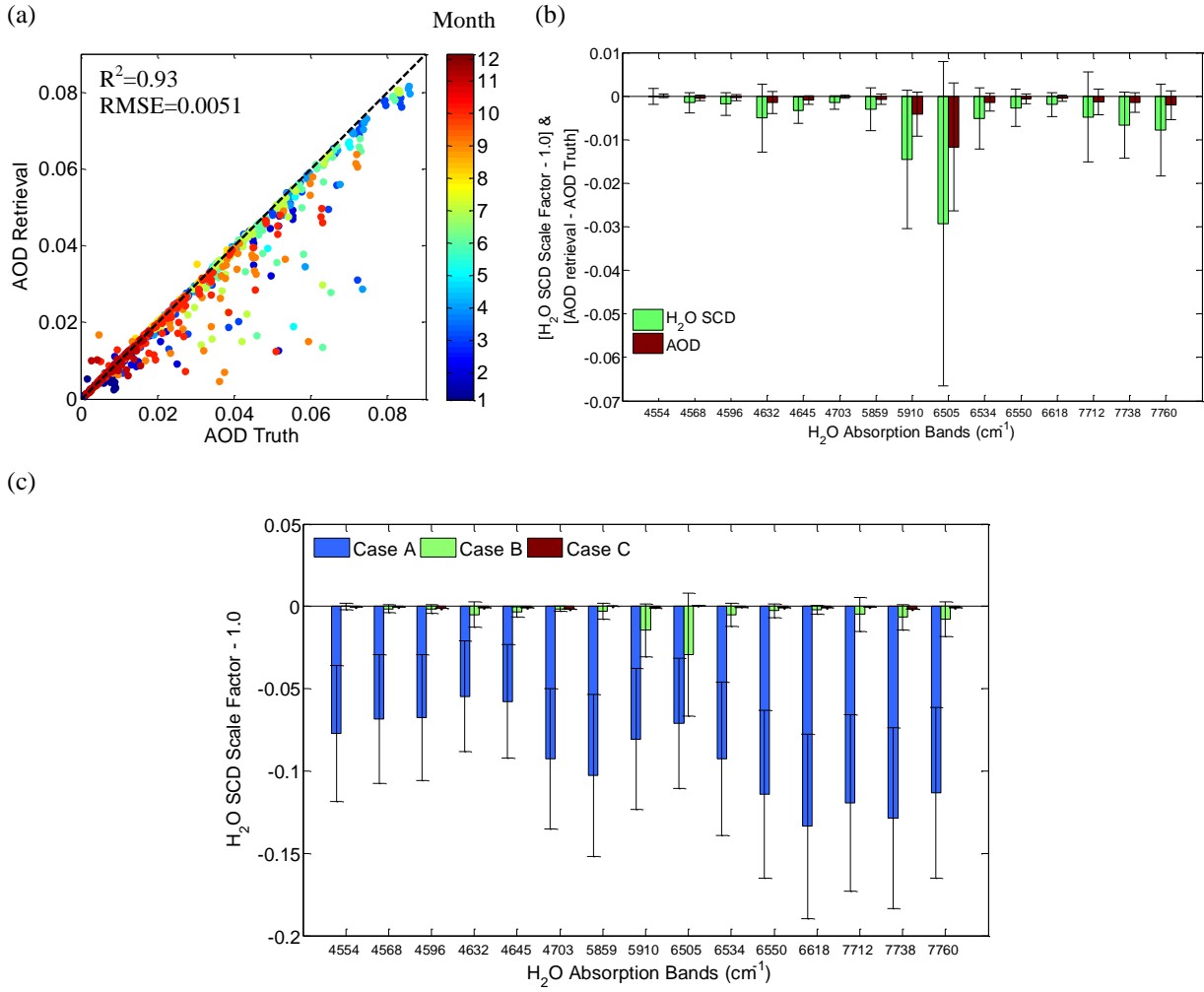

**Figure 8**. Retrieval of AOD and $H_2O$ SCD simultaneously based on a realistic numerical simulation study using the 2S-ESS RT model. (a) Scatter plots between true and retrieved AOD from the simultaneous retrieval experiment. The mean and standard deviation of the difference between them are −0.0018 and 0.0051, respectively. The black dotted line is the one-to-one line; (b) Retrieval of AOD and $H_2O$ SCD scale factors averaged over the 15 $H_2O$ absorption bands from the simultaneous retrieval experiment; the one sigma error bar is also shown; (c) $H_2O$ SCD scale factor retrievals from three different cases; in Case A, aerosol scattering is not considered; in Case B, $H_2O$ and AOD are simultaneously retrieved; in Case C, AOD is perfectly known.

*Supplement of*

# Aerosol Scattering Effects on Water Vapor Retrievals over the Los Angeles Basin

Zhao-Cheng Zeng et al.

*Correspondence to*: zcz@gps.caltech.edu

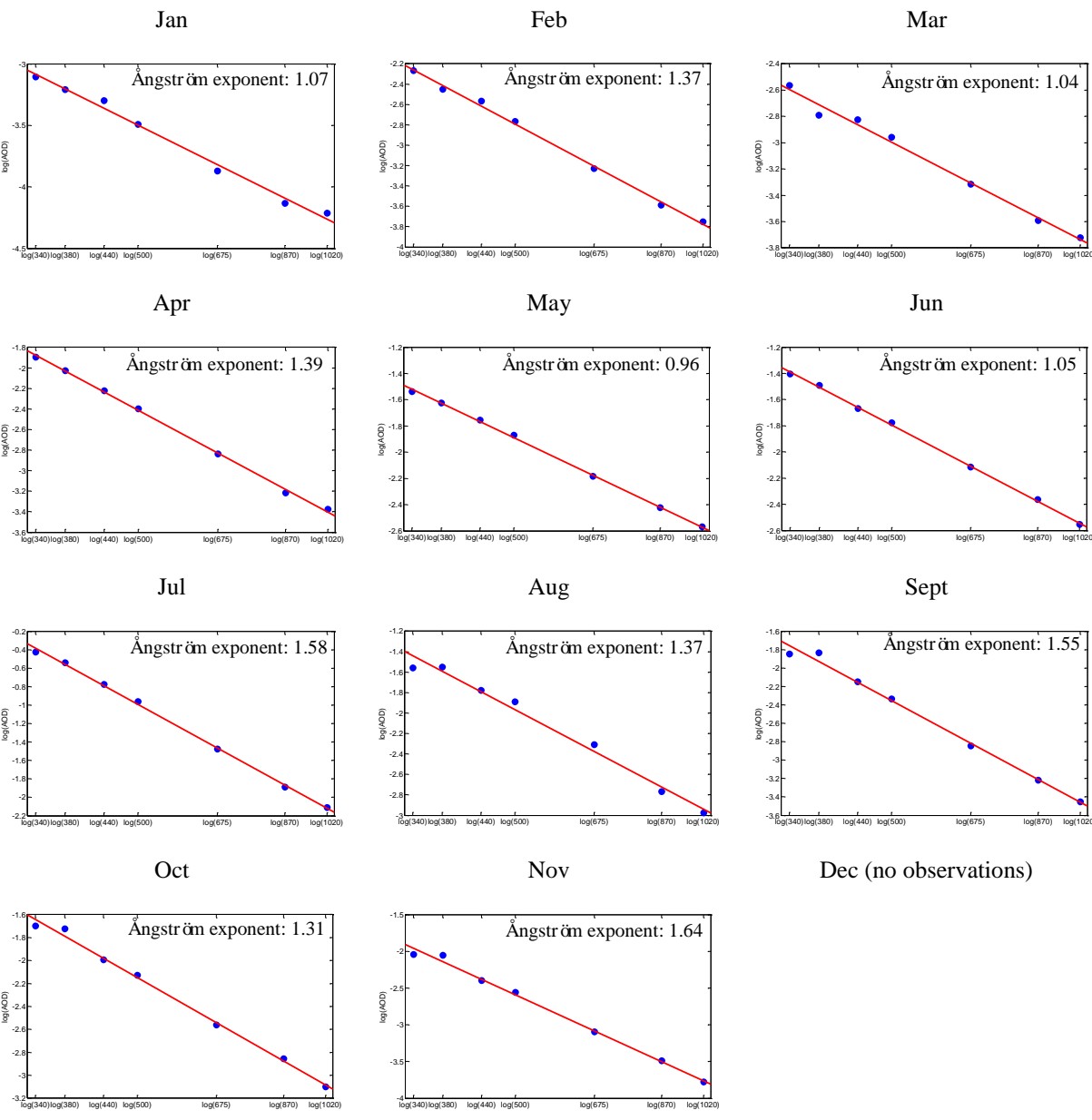

**Supplemental Material Figure 1**. Examples from applying linear regression, using the logarithmic form of Ångström exponent law, on the AERONET AOD data at noon time around 13:00, for the 11 different months, selected from the 68 days of data shown in Figure 4 and 7. The blue dots are the AERONET AOD measurements in the seven different bands (340 nm, 380 nm, 440 nm, 500 nm, 870 nm and 1020 nm) and the red line is the linear regression result. Angstrom exponents are also indicated. Please note that both the x- and y-axes is in logarithmic scale.

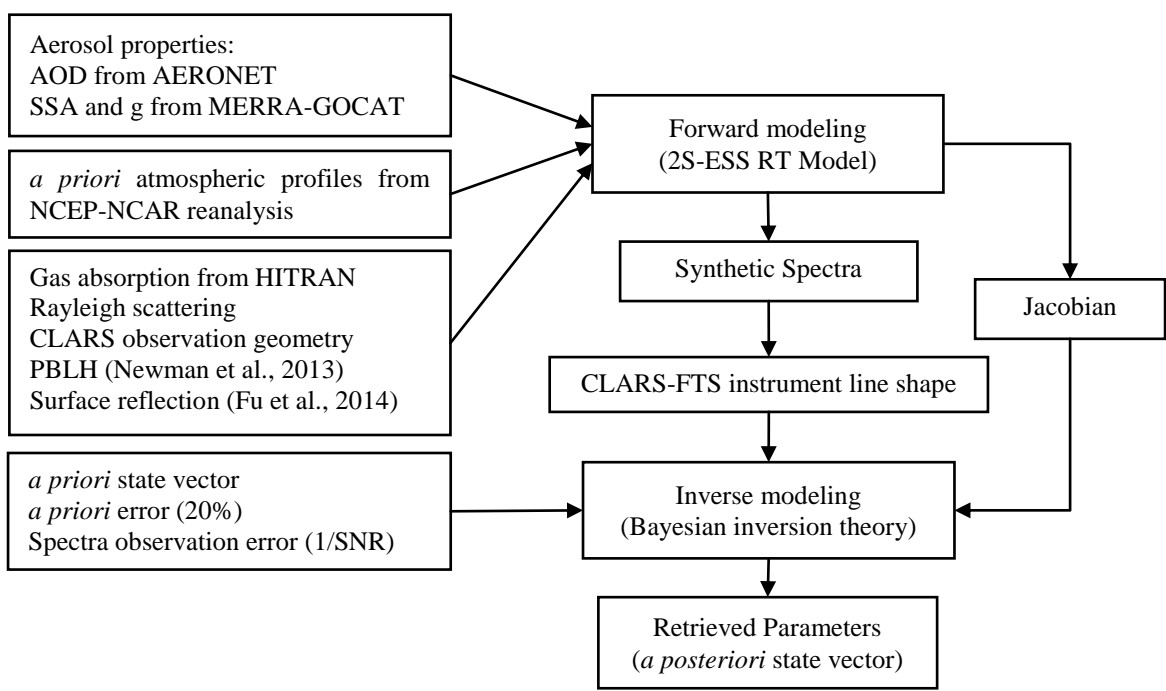

**Supplementary Material Figure 2.** Schematic diagram of the retrieval algorithm based on the 2S-ESS RT model and Bayesian inversion theory. A detailed description is provided in Section 5.

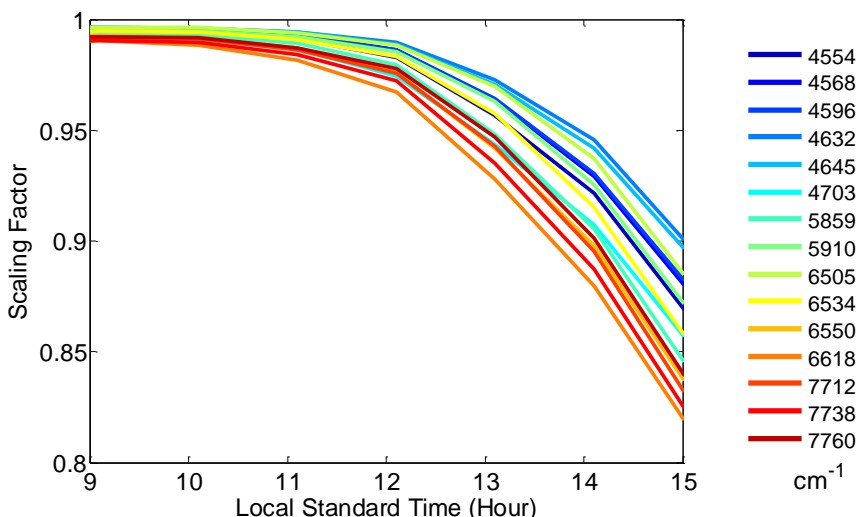

**Supplementary Material Figure 3**. Scaling factors for H$_2$O SCDs retrieved from the simulated synthetic spectral radiance in the 15 chosen bands using 2S-ESS RT model with AOD data from AERONET-Caltech on March 01, 2013. This plot is the same as Figure 6(a), except that the mean is not subtracted.

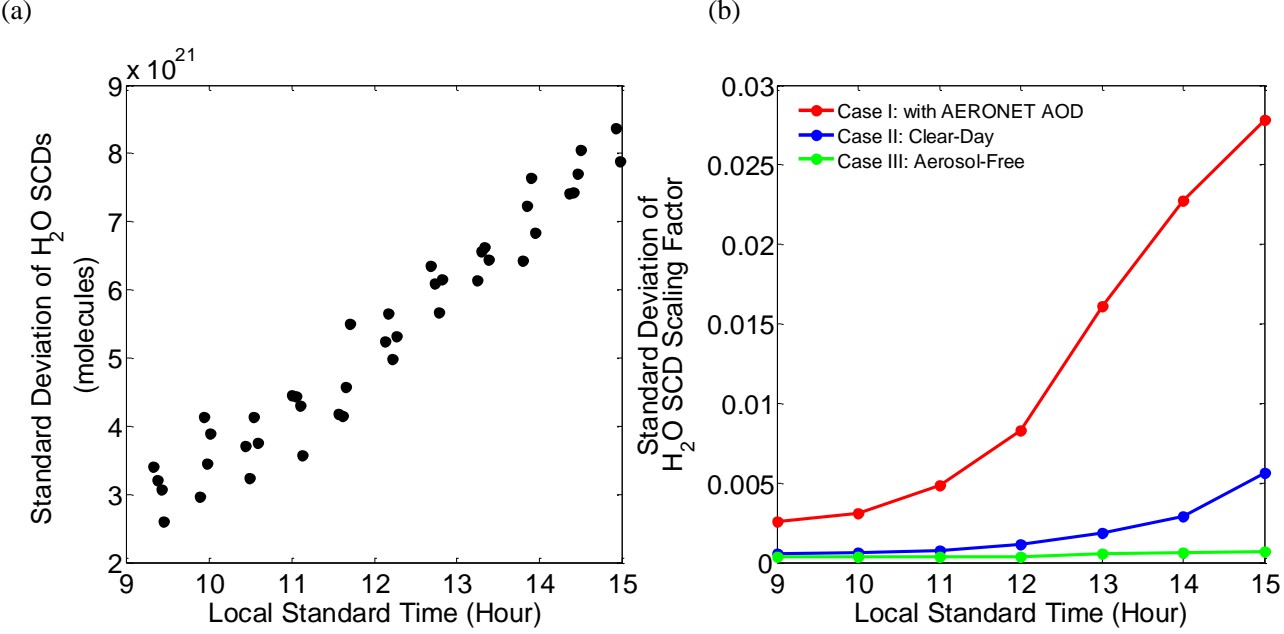

**Supplementary Material Figure 4**. Standard deviation of $H_2O$ (a) SCDs observed by CLARS (unit: molecules), and (b) SCD scaling factors simulated by the 2S-ESS RT model. The three cases in (b) are the same as those in Figure (6).

