# Peer review of "Aerosol Scattering Effects on Water Vapor Retrievals over the Los Angeles Basin"

_Atmospheric Chemistry and Physics, 2016_

## Referee Comment (RC1) · Anonymous Referee #1 · 20 Jul 2016

General comments:

The authors describe a potential method that can assist the derivation of aerosol effects of GHG retrievals from space. The method uses measurements and retrievals of water vapor from lines at the SWIR (1250-2500nm), and which shows dependency in water vapor retrievals that are correlated with the amount of aerosols in the atmosphere. The overall idea is innovative and interesting. However, there are some caveats to the method, and some of the assumptions that the method rely on, are not completely supported in the manuscript. The manuscript would benefit from additional sensitivity

tests (as described below) and should emphasize or describe more accurately how the suggested method can assist in constraining aerosol optical properties and which optical properties exactly. Overall, it is not entirely clear how the method can assist real-time retrievals, since the method described uses pre-assigned aerosol properties and phase function to get the best fit of the water vapor. The authors did not discuss how different type and aerosol properties might affect that goodness of fit. It seems that the aerosol amount (i.e., AOD), and not the specific aerosol property are responsible for the trend in the water vapor. Hence, it is not clear how much the method is sensitive or helpful to constrain aerosol optical properties (e.g. size distribution, SSA, refractive index), as stated, or just aerosol relative amount/trends. It would be insightful to see how simulations using different aerosol properties as input, scaled by the AERONET AOD during the day, changes the best fit, as shown in Fig. 5c. Also, the authors state that their analysis is based on water vapor absorption alone in the wavelength range of investigation. However, $CO_2$, $CH_4$ and $N_2O$ absorb as well in this wavelength range and the authors should mention how the combined absorption of those gases might affect the suggested method in terms of the IC, and the ability to de-convolve the aerosol trend from the GHG mixture, given their variability during the day. Given a situation where aerosol amount/properties are unknown (i.e. real retrieval), what would be the uncertainty level of the retrieved water vapor amount and aerosol (properties or AOD?). The authors are showing results from wavelengths in the range of 1250-2500 nm, while the AERONET instrument measure only up to 1020 nm. The authors state that they use an extrapolated angstrom exponent to derive the aerosol wavelength dependency to use in the model, however, at the FTS wavelength range, there is already very little dependence, as already between 870 and 1020 nm bands there is very little dependence (Fig. d, both panels). Hence, it is unclear how the method can constrain the aerosol properties since in this wavelength range the wavelength dependency is very weak. What is the IC available for constraining aerosol properties? Again, it seems that the method can give the general aerosol amount, but cannot differentiate between different aerosol types, which have different optical properties. Also, from Fig. 4, it

is interesting to note that the aerosol wavelength dependency is changing during the course of the day. Did the authors explored how this local behavior might affect the suggested method?

Some additional questions, and points to note:

1. Small aerosol, such as urban pollution and Biomass burning are not expected to have such a large scattering effect at the FTS wavelength range. Please expand the discussion on this and on the ability of the method to be helpful under events that are dominated by these type of aerosols, rather than dust for example. 2. The authors are stating that the method can assist in constraining aerosol optical properties, but the majority of the discussion is around AOD, which is not an internal property of the aerosol. Please try to define the objectives and discussion in a more accurate way. 3. Please provide an explanation of the GHG retrievals, especially on the spectral range of interest and whether these are overlapping with the wavelength range of the water vapor measurements. How these would interfere with each other in an end-to-end retrieval scheme?

Minor comments: Fig. 3, please add label on the x-axis Page 6, lines 13-15, it is not clear whether the RT simulations are being done for 5 aerosol type or a combination of those 5 to give a mixture aerosol type that should represent the LA basin aerosols.

---

## Referee Comment (RC2) · Anonymous Referee #2 · 22 Jul 2016

Review of "Investigating Wavelength-Dependent Aerosol Optical Properties Using Water Vapor Slant Column Retrievals from CLARS over the Los Angeles Basin" by Zeng et al.

This paper discusses Mt. Wilson based CLARS measurements of $H_2O$ slant column density in the 4000 – 8000 cm-1 range. The standard deviation of the $H_2O$ slant column, retrieved in 15 spectral bands, based upon model fits that exclude aerosol, is then compared to an AERONET aerosol optical path (AOP) value. Scatter diagrams of $H_2O$ standard deviation versus AOP for a variety of dates have a correlation R2 near

0.5.

Though the scatter diagrams indicate that the model-fit residuals do indicate the presence of aerosol, the paper as it now stands does not calculate wavelength-dependent aerosol optical properties. Aerosol optical properties, from my perspective, refers to such quantities such as aerosol optical depth, size distribution, real and imaginary refractive indices, etc. When I read the title "Investigating Wavelength-Dependent Aerosol Optical Properties.." I assumed that actual properties (at the very least, aerosol optical depths) would be retrieved from the CLARS measurements. The paper does "suggest that wavelength-dependent aerosol optical properties can be constrained", but I feel the "suggestion" stage does not in itself go far enough.

Further work, in which optical properties from CLARS retrievals and AERONET observations are compared, is recommended before publication of this paper.

Other comments

Page 3, lines 10-11. The sentence "It is worth noting.." is not clear. Does this refer to other studies which have analyzed CLARS-FTS measurements, or processing that refers to the paper's calculations?

Page 3, line 26. "They reflect the precision of the H2O"

Page 4, line 25. "and changes in the spatial distribution of the aerosol".

Page 6, line 24. Is the AERONET-Caltech measurements of aerosol optical depth insensitive to H2O (e.g. the H2O band wavelengths need not be included in the AERONET retrieval)?

Page 6, line 24. How is the Ängström exponent chosen? Please clarify.

Page 7, line 31. In regard to the 5 km distance between West Pasadena and Caltech, why is there not a target installed on the Caltech campus? Is the CLARS observatory to Caltech campus line of sight not possible? Line 16 on page 8 refers to other future

targets. If Caltech is included in the expanded list of future targets, please mention this.

Page 8, line 1. "Caltech also represents that in West Pasadena."

Page 8, line 19. The claim "We illustrate the robust ability of multi-wavelength retrievals of water vapor slant columns to provide constraints on aerosol optical properties" is not demonstrated by the current paper since optical properties are not retrieved.

Page 8, line 28. The proposed approach is proposed, not demonstrated, in regards to providing "a sensitive way to quantify the effect of aerosol scattering in GHG retrievals". I encourage some additional work that supports this assertion.

Please also note the supplement to this comment:
http://www.atmos-chem-phys-discuss.net/acp-2016-490/acp-2016-490-RC2-supplement.pdf

---

## Author Comment (AC1) · 28 Oct 2016

We have made major and careful modifications to the original manuscript according to all the comments and suggestions from the reviewers. The three major modifications include:

(1) Clarification of the objective of this study

The primary aim of this study is to propose a novel, potential approach to describe the wavelength-dependent aerosol scattering effects using $H_2O$ retrievals, rather than to only retrieve any specific aerosol optical properties as we stated in our first draft, but a series of sensitivity tests to investigate the impacts from aerosol scattering properties on our results have been added in this revised manuscript. This study is an important step towards the direction of fully quantifying the aerosol properties and aerosol scattering effects relevant to greenhouse gas (GHG) retrievals from space. We demonstrated the proposed approach in a complex urban environment in the LA basin using measurements from CLARS and simulations from a 2S-ESS radiative transfer (RT) model. The results from this study show that (1) aerosol scattering effect is the primary contributor to the variations in $H_2O$ SCDs retrieved from multiple bands, and (2) a significant linear correlation is also found between variations in $H_2O$ SCD retrievals from multiple bands and corresponding AOD data; this correlation is associated with asymmetry parameter (g), which is a first-order measure of the aerosol scattering phase function. The conclusion is that, these evidences from both measurements and simulations suggest that wavelength-dependent aerosol scattering effects can be derived using $H_2O$ retrievals from multiple bands. In the revised manuscript (attached in the end), we changed the title and abstract accordingly.

(2) Time range and control experiments in Figure 6

We changed the time range (Figure 3 and 6) to between 9:00 and 15:00 local time to show only the data with solar zenith angles (SZAs) less than 60°, similar to the majority of satellite observations, to exclude the retrievals with large SZAs which may introduce large uncertainty. This time range includes the local overpass times of around 10:00 for SCIAMCHY and around 13:30 for GOSAT, OCO-2, and upcoming TanSat. To explore the role of aerosol scattering in the variations in $H_2O$ SCDs, two new control experiments are shown in Figure 6. In the first control experiment, the AOD data are fixed at the clear-day level, for which the lowest AOD across the year is used, for all hours across the day. In the second control experiment, the AOD is fixed at the aerosol-free level, which is the zero for all hours across the day.

(3) Sensitivity tests on aerosol scattering properties

Figure 7 is added to examine the sensitivity of the correlation, between variations in $H_2O$ SCDs and the corresponding AOP, to aerosol scattering properties, mainly single scattering albedo (SSA) and asymmetry parameter (g), using the 2S-ESS RT model with aerosol compositions from five composite types. Four different control experiments are implemented. The results show that (a) correlation between AOD and extent of variations in $H_2O$ SCDs is robust and significant, and the difference in $H_2O$ abundance and g are responsible for the difference of slopes between the winter-spring and summer-autumn periods; (b) the variations of SSA and g are not the key contributors to the variations of the standard deviations of $H_2O$ SCDs; and (c) g is the main contributor to the ratio of the changes (the slope from linear regression). Moreover, Figure 5 was added to show the monthly-averaged climatological aerosol compositions for the five composite MERRA aerosols in the basin, and Table 1 was added to show the corresponding aerosol single scattering properties for the five composite aerosols.

Please refer to the revised manuscript (attached in the end) for details.

Anonymous Referee #1

We thank the Reviewer #1 for his/her constructive comments and suggestions to improve the quality and clarity of our manuscript.

5    Item-by-item responses to the specific comments are provided below, in which the reviews' comments are in **blue**, our responses in **black**, and modifications of the original manuscript are indicated by highlight in **yellow**. The revised manuscript with highlighted changes is attached in the end.

**General comments:**

10   The authors describe a potential method that can assist the derivation of aerosol effects of GHG retrievals from space. The method uses measurements and retrievals of water vapor from lines at the SWIR (1250-2500nm), and which shows dependency in water vapor retrievals that are correlated with the amount of aerosols in the atmosphere. The overall idea is innovative and interesting. However, there are some caveats to the method, and some of the assumptions that the method rely on, are not completely supported in the

15   manuscript. The manuscript would benefit from additional sensitivity tests (as described below) and should emphasize or describe more accurately how the suggested method can assist in constraining aerosol optical properties and which optical properties exactly. Overall, it is not entirely clear how the method can assist real-time retrievals, since the method described uses pre-assigned aerosol properties and phase function to get the best fit of the water vapor.

20   We thank the reviewer for this comment which led us to make substantial revisions to the manuscript. In the revised manuscript, we clarified the aim of this study. It is to propose a novel approach to describe the wavelength-dependent aerosol scattering effects using $H_2O$ retrievals from multiple bands. We also added Figure 7 to examine the sensitivity of the linear correlation between AOD and the variations in $H_2O$ SCDs on SSA and g, two important scattering properties of aerosol. Rather than providing a new method of real-

25   time retrieval of AOD or other optical properties, our proposed approach improves our understanding of aerosol scattering effects on $H_2O$ retrievals, which provides a sensitive way to quantify the effects of aerosol scattering in greenhouse gas retrievals and potentially contribute toward reducing errors of greenhouse gas retrievals from space.

30   The authors did not discuss how different type and aerosol properties might affect that goodness of fit. It seems that the aerosol amount (i.e., AOD), and not the specific aerosol property are responsible for the trend in the water vapor. Hence, it is not clear how much the method is sensitive or helpful to constrain aerosol optical properties (e.g. size distribution, SSA, refractive index), as stated, or just aerosol relative amount/trends. It would be insightful to see how simulations using different aerosol properties as input,

35   scaled by the AERONET AOD during the day, changes the best fit, as shown in Fig. 5c.

We appreciate these suggestions. As shown in Figure 4, AOD is found to linearly correlated with variations in $H_2O$ SCDs. To examine the sensitivity of the correlation to aerosol scattering properties, mainly SSA and g, we run the 2S-ESS RT model, with AOD inputs from AERONET, daily varying aerosol compositions (five types: organic carbon, black carbon, sulfate, dust, sea salt) from MERRA reanalysis data

40   and single scattering aerosol properties calculated by the GOCART model. The result is shown in Figure 7 in the revised manuscript. From the control experiments, we found that (a) correlation between AOD and

variations in $H_2O$ SCDs is robust and significant, and the difference in $H_2O$ abundance and g are responsible for the difference of slopes between the winter-spring and summer-autumn periods; (b) the variations of SSA and g are not the key contributors to the variations of the standard deviations of $H_2O$ SCDs; and (c) g is the main contributor to the ratio of AOD changes to variations in the retrieved $H_2O$ SCD. For more detail, please see Section 5.2 in the revised manuscript.

Also, the authors state that their analysis is based on water vapor absorption alone in the wavelength range of investigation. However, CO2, CH4 and N2O absorb as well in this wavelength range and the authors should mention how the combined absorption of those gases might affect the suggested method in terms of the IC, and the ability to de-convolve the aerosol trend from the GHG mixture, given their variability during the day.

Thank you for this comment, pointing out that clarification was needed. Absorptions by other gas molecules have been considered in the 2S-ESS RT model used. In the model, absorptions by the dominant gas molecules in the atmosphere, including $H_2O$, $CO_2$, $N_2O$, CO, $CH_4$, $O_2$, $N_2$ and HDO, are considered by using an *a priori* atmospheric profile obtained from the National Centers for Environmental Prediction (NCEP)-National Center for Atmospheric Research (NCAR) reanalysis data (Kalnay et al., 1996). We added the above statements to Section 3 in the revised manuscript.

In the chosen 15 $H_2O$ absorption bands, $H_2O$ absorption is dominant. The ratio of absorption by water vapor to the all gases for the 15 bands is greater than 90% on average. Given the dominant absorption of $H_2O$ in these bands, full consideration of the absorption by other gases, and the high IC of retrieving $H_2O$ SCDs, we believe that the retrievals of $H_2O$ SCDs from the 15 bands should be accurate and the impact from other gases on the relation between AOD and variation in $H_2O$ should be negligible.

Given a situation where aerosol amount/properties are unknown (i.e. real retrieval), what would be the uncertainty level of the retrieved water vapor amount and aerosol (properties or AOD?).

Thank you for this question. As shown in Figure 7 of the revised manuscript, the variations in $H_2O$ SCDs are robustly and linearly correlated with AOD, and the rate of change is related to the asymmetry parameter of aerosol scattering. The coefficient of determination from linear regression can be used to indicate the uncertainty level. From the CLARS measurements, the AOD variability can explain about 50% of the variations of scattering effects on $H_2O$ retrievals (Figure 4), while based on the model simulations from the 2S-ESS model, AOD variability can explain about 76-80% (Figure 7(a)).

The authors are showing results from wavelengths in the range of 1250-2500 nm, while the AERONET instrument measure only up to 1020 nm. The authors state that they use an extrapolated angstrom exponent to derive the aerosol wavelength dependency to use in the model, however, at the FTS wavelength range, there is already very little dependence, as already between 870 and 1020 nm bands there is very little dependence (Fig. d, both panels). Hence, it is unclear how the method can constrain the aerosol properties since in this wavelength range the wavelength dependency is very weak.

[Figure]

**Figure I**. examples from applying linear regressions using the logarithmic form of Ångström exponent law on the AERONET AOD data at noon time around 13:00, for the 11 different months, selected from the 68 days of data shown in Figure 4 and 7. The blue dots are the AERONET AOD measurements in the seven different bands (340nm, 380nm, 440nm, 500nm, 870nm and 1020nm) and the red line is linear regression result from the first 6 of the 7 bands (that is, 1020nm is excluded). Therefore, the closeness of the AOD at 1020nm to the fitted red line indicates the strength of the wavelength dependence that follows the exponential law. The averaged bias between the AOD and the red line value at 1020nm for all the 68 days is 9.82±10.02% (mean and one standard deviation).

Thank you for this comment. To examine the wavelength dependence of all the data used, especially on the near-infrared band at 1020nm, we checked the AERONET AOD data that were used in Figure 4 and Figure 7 with 68 days of data. The dependence can be described by the following Ångström exponent law:

$$\frac{\tau}{\tau_0} = (\frac{\lambda}{\lambda_0})^{-k} \quad \textbf{OR} \quad \log(\tau) = -k * \log(\lambda) + c \ [c \ is \ a \ constant]$$

Where $\lambda$ and $\tau$ are the wavelength and the corresponding AERONET AOD, and $\lambda_0$ and $\tau_0$ are the reference wavelength and the corresponding AOD, and $k$ is the Ångström exponent. If the AOD measurements in different wavelengths follow this exponent law, we would expect the AOD has strong wavelength dependence. Figure I shows examples from applying linear regressions using the logarithmic form of Ångström exponent law on the AERONET AOD data at noon time around 13:00, for the 11 different months, selected from the 68 days of data shown in Figure 4 and 7. The blue dots are the AERONET AOD

measurements in the seven different bands (340nm, 380nm, 440nm, 500nm, 870nm and 1020nm) and the red line is linear regression result from the first 6 of the 7 bands (that is, 1020nm is excluded). Therefore, the closeness of the AOD at 1020nm to the fitted red line indicates the strength of the wavelength dependence that follows the exponential law. We can see, in general, the wavelength dependence follows the exponent law, suggesting the dependence is strong. We further calculated the averaged bias between the AOD and the red line value at 1020nm. The average bias for all the 68 days is 9.82±10.02% (mean and one standard deviation), which is small. Therefore, we conclude that the wavelength dependences for most of the 68 days are strong. We added Figure I in the supplementary material.

What is the IC available for constraining aerosol properties?

Thank you for this question. In this study, we used the IC value to evaluate the efficiency of band channels selected for retrieval and assess the precision of $H_2O$ retrieval in the selected 15 bands. However, it is hard to relate the IC values with the correlation between AOD (or other aerosol scattering properties) and variations in $H_2O$ SCDs. Instead, we used the $R^2$ quantify the uncertainty of the correlation between them, and control experiments using 2S-ESS to examine its sensitivity to aerosol scattering properties. From the CLARS measurements, the AOD variability can explain about 50% of the variations of scattering effects on $H_2O$ retrievals (Figure 4), while based on the model simulations from the 2S-ESS model, AOD variability can explain about 76-80% (Figure 7(a)). The results from sensitivity test is presented in Section 5.2 in the revised manuscript.

Again, it seems that the method can give the general aerosol amount, but cannot differentiate between different aerosol types, which have different optical properties.

Thank you for this comment. We agree that we do not differentiate between different aerosol types. One of the key conclusions from this study is that AOD (from all aerosol types) is linearly and significantly correlated with variations in H2O SCDs, and that the asymmetric parameter, relevant to aerosol scattering phase, is responsible for the rate of changes between them. However, we cannot differentiate between different aerosol types. But in the revised manuscript, we added Figure 7 to examine the sensitivity of this linear correlation on SSA and g, two important scattering properties of aerosol that are influenced by the aerosol compositions (five composition types from MERRA were used: organic carbon, black carbon, sulfate, dust and sea salt) and their different scattering properties. Please refer to Section 5.2 in the revised manuscript for details.

Also, from Fig. 4, it is interesting to note that the aerosol wavelength dependency is changing during the course of the day. Did the authors explored how this local behavior might affect the suggested method?

Thank you for this comment. The changing wavelength dependency of AOD can be inferred from the changing Ångström parameter during the day, as shown in the following figure II, and this change of Ångström parameter results from the change of aerosol compositions. In Figure 7 of the revised manuscript, we show results of four controlled sensitivity tests using the aerosol compositions from MERRA reanalysis data for five composite types (organic carbon, black carbon, sulfate, dust, sea salt) using the 2S-ESS model. The results show that the effective asymmetry parameter of aerosol scattering, as a composition-weighted sum, has impact on the correlation slope between AOD and the magnitude of the variations in $H_2O$ SCDs. Please refer to Section 5.2 in the revised manuscript for more details.

[Figure]

**Figure II**. Angstrom parameter from AEROENT-Caltech on March 01, 2013. This figure was downloaded from http://aeronet.gsfc.nasa.gov/.

**Some additional questions, and points to note:**

1. Small aerosol, such as urban pollution and Biomass burning are not expected to have such a large scattering effect at the FTS wavelength range. Please expand the discussion on this and on the ability of the method to be helpful under events that are dominated by these type of aerosols, rather than dust for example.

Thank you for this good suggestion. Figure 5 was added in the revised manuscript to show the monthly averaged aerosol compositions (five composition types from MERRA: organic carbon, black carbon, sulfate, dust and sea salt) in 2013 in the LA basin, and their corresponding aerosol scattering properties are also shown in Table 1. We can see that, in general, sea salt dominates in the summer-autumn period while dust dominates in the winter-spring period. For small aerosols, such as black carbon, organic carbon and sulfate, their ratios are relatively small but changes during the year with larger proportion in summer than in winter. From Table 1 we can see that the SSA and g of these small aerosols are much smaller than that of dust and sea salt. Therefore, under the events that dominate by these types of small aerosols, the effective g will become smaller and therefore the rate of change (the slope from linear regression) will become smaller, according to the results from control experiments shown in Figure 7(e). We added these statements in the Section 5.2 of the revised manuscript.

2. The authors are stating that the method can assist in constraining aerosol optical properties, but the majority of the discussion is around AOD, which is not an internal property of the aerosol. Please try to define the objectives and discussion in a more accurate way.

Thank you for this good comment which led us to make substantial revisions to the manuscript. In the revised manuscript, we clarified the aim of this study. It is to propose a novel approach to describe the wavelength-dependent aerosol scattering effects using $H_2O$ retrievals from multiple bands. We also added Figure 7 to examine the sensitivity of the linear correlation between AOD and the variations in $H_2O$ SCDs

on SSA and g, two important scattering properties of aerosol. We also changed the title and abstract accordingly.

3. Please provide an explanation of the GHG retrievals, especially on the spectral range of interest and whether these are overlapping with the wavelength range of the water vapor measurements. How these would interfere with each other in an end-to-end retrieval scheme?

The 15 bands we chosen to retrieve $H_2O$ lie in the range of 4000 to 8000 $cm^{-1}$. The GHGs, such as $CO_2$ and $CH_4$, are retrieved from current satellite missions using GHGs absorption bands in around 6200 $cm^{-1}$ (weak $CO_2$ absorption band) and 4850 $cm^{-1}$ (strong $CO_2$ absorption band) for $CO_2$ and 6000 $cm^{-1}$ for $CH_4$.

In our study, the 15 $H_2O$ bands are dominated by $H_2O$ absorptions. Apart from $H_2O$, in the 2S-ESS model, absorptions by other gas molecules in the atmosphere, including $CO_2$, $N_2O$, CO, $CH_4$, $O_2$, $N_2$ and HDO, are considered by using an *a priori* atmospheric profile obtained from NCEP-NCAR reanalysis data. Even though in this study, we assume that only one state variable, i.e., $H_2O$ SCD, is retrieved. But from the IC results shown in Figure 2, we can see that the average IC is high (about 6 on average) and similar among all bands, which indicates that our retrieved $H_2O$ SCD has very high retrieval precision compared with the *a priori* information.

Retrieval of GHGs from satellite usually retrieves the GHGs and $H_2O$ simultaneously to account for the impact from $H_2O$ absorption, which is nonnegligible in the $CO_2$ or $CH_4$ absorption bands.

**Minor comments:**

Fig. 3, please add label on the x-axis

Done.

Page 6, lines 13-15, it is not clear whether the RT simulations are being done for 5 aerosol type or a combination of those 5 to give a mixture aerosol type that should represent the LA basin aerosols.

In this study, the RT simulations were done for a combination of the five composite aerosol types in the LA basin. Using the compositions of the five composite MERRA aerosols and their scattering properties, the effective SSA and g of aerosol scattering are calculated, respectively, as the composition-weighted sum for all types, and then incorporated into the 2S-ESS RT model. We added the above statements to Section 5 in the revised manuscript.

**The following documents are the revised manuscript (with changes highlighted) and the supplementary figure:**

[revised manuscript text omitted]
) and the red line is linear regression result from the first 6 of the 7 bands (that is, 1020nm is excluded). Therefore, the closeness of the AOD at 1020nm to the fitted red line indicates the strength of the wavelength dependence that follows the exponential law. The averaged bias between the AOD and the red line value at 1020nm for all the 68 days is 9.82±10.02% (mean and one standard deviation).

---

## Author Comment (AC2) · 28 Oct 2016

We thank the Reviewer#2 for his/her constructive comments and suggestions to improve the quality and clarity of our manuscript.

Item-to-item responses to the specific comments are provided below, in which the reviews' comments are in **blue**, our responses in **black**, and modifications of the original manuscript are indicated by highlight in **yellow**. The revised manuscript with highlighted changes is attached in the end.

Review of "Investigating Wavelength-Dependent Aerosol Optical Properties Using Water Vapor Slant Column Retrievals from CLARS over the Los Angeles Basin" by Zeng et al.

This paper discusses Mt. Wilson based CLARS measurements of H2O slant column density in the 4000 – 8000 cm-1 range. The standard deviation of the H2O slant column, retrieved in 15 spectral bands, based upon model fits that exclude aerosol, is then compared to an AERONET aerosol optical path (AOP) value. Scatter diagrams of H2O standard deviation versus AOP for a variety of dates have a correlation R2 near 0.5. Though the scatter diagrams indicate that the model-fit residuals do indicate the presence of aerosol, the paper as it now stands does not calculate wavelength-dependent aerosol optical properties. Aerosol optical properties, from my perspective, refers to such quantities such as aerosol optical depth, size distribution, real and imaginary refractive indices, etc. When I read the title "Investigating Wavelength-Dependent Aerosol Optical Properties.." I assumed that actual properties (at the very least, aerosol optical depths) would be retrieved from the CLARS measurements. The paper does "suggest that wavelength-dependent aerosol optical properties can be constrained", but I feel the "suggestion" stage does not in itself go far enough. Further work, in which optical properties from CLARS retrievals and AERONET observations are compared, is recommended before publication of this paper.

We thank the reviewer for this comment which led us to make substantial revisions to the manuscript. The three major modifications include:

(1) Clarification of the objective of this study

The primary aim of this study is to propose a novel, potential approach to describe the wavelength-dependent aerosol scattering effects using $H_2O$ retrievals, rather than to only retrieve any specific aerosol optical properties as we stated in our first draft, but a series of sensitivity tests to investigate the impacts from aerosol scattering properties on our results have been added in this revised manuscript. This study is an important step towards the direction of fully quantifying the aerosol properties and aerosol scattering effects relevant to greenhouse gas (GHG) retrievals from space. We demonstrated the proposed approach in a complex urban environment in the LA basin using measurements from CLARS and simulations from a 2S-ESS radiative transfer (RT) model. The results from this study show that (1) aerosol scattering effect is the primary contributor to the variations in $H_2O$ SCDs retrieved from multiple bands, and (2) a significant linear correlation is also found between variations in $H_2O$ SCD retrievals from multiple bands and corresponding AOD data; this correlation is associated with asymmetry parameter (g), which is a first-order measure of the aerosol scattering phase function. The conclusion is that, these evidences from both measurements and simulations suggest that wavelength-dependent aerosol scattering effects can be derived using $H_2O$ retrievals from multiple bands. In the revised manuscript (attached in the end), we changed the title and abstract accordingly.

(2) Time range and control experiments in Figure 6

We changed the time range (Figure 3 and 6) to between 9:00 and 15:00 local time to show only the data with solar zenith angles (SZAs) less than 60°, similar to the majority of satellite observations, to exclude the retrievals with large SZAs which may introduce large uncertainty. This time range includes the local overpass times of around 10:00 for SCIAMCHY and around 13:30 for GOSAT, OCO-2, and upcoming TanSat. To explore the role of aerosol scattering in the variations in $H_2O$ SCDs, two new control experiments are shown in Figure 6. In the first control experiment, the AOD data are fixed at the clear-day level, for which the lowest AOD across the year is used, for all hours across the day. In the second control experiment, the AOD is fixed at the aerosol-free level, which is the zero for all hours across the day.

(3) Sensitivity tests on aerosol scattering properties

Figure 7 is added to examine the sensitivity of the correlation, between variations in $H_2O$ SCDs and the corresponding AOP, to aerosol scattering properties, mainly single scattering albedo (SSA) and asymmetry parameter (g), using the 2S-ESS RT model with aerosol compositions from five composite types. Four different control experiments are implemented. The results show that (a) correlation between AOD and extent of variations in $H_2O$ SCDs is robust and significant, and the difference in $H_2O$ abundance and g are responsible for the difference of slopes between the winter-spring and summer-autumn periods; (b) the variations of SSA and g are not the key contributors to the variations of the standard deviations of $H_2O$ SCDs; and (c) g is the main contributor to the ratio of the changes (the slope from linear regression). Moreover, Figure 5 was added to show the monthly-averaged climatological aerosol compositions for the five composite MERRA aerosols in the basin, and Table 1 was added to show the corresponding aerosol single scattering properties for the five composite aerosols.

Please refer to the revised manuscript (attached in the end) for details.

**Other comments**

Page 3, lines 10-11. The sentence "It is worth noting.." is not clear. Does this refer to other studies which have analyzed CLARS-FTS measurements, or processing that refers to the paper's calculations?

This sentence describes the current products of XGHG from CLARS. In the revised manuscript. we deleted this sentence to avoid ambiguity, since it is not relevant to the results from this study.

Page 3, line 26. "They reflect the precision of the H2O"

We rephrased the sentence to "They are indicators of the precision of $H_2O$ retrieval in the selected bands".

Page 4, line 25. "and changes in the spatial distribution of the aerosol".

This was removed in the revised manuscript since it is not relevant.

Page 6, line 24. Is the AERONET-Caltech measurements of aerosol optical depth insensitive to H2O (e.g. the H2O band wavelengths need not be included in the AERONET retrieval)?

Thank you for this comment, and sorry for the misunderstanding and ambiguity it caused. We rephrased the sentence to: "however, the wavelengths of the 15 $H_2O$ absorption bands used in this study, ranging from about 1280 nm to 2200 nm, are outside the AERONET wavelength range."

5     Page 6, line 24. How is the Ängström exponent chosen? Please clarify.

The Ängström exponent law is given by,

$$\frac{\tau}{\tau_0} = (\frac{\lambda}{\lambda_0})^{-k}$$

or by its logarithmic form,

$$\log(\tau) = -k * \log(\lambda) + c \ [c \ is \ a \ constant]$$

10     where $\lambda$ and $\tau$ are the wavelength and corresponding AERONET AOD measurement, and $\lambda_0$ and $\tau_0$ are the reference wavelength and the corresponding AOD, and $k$ is the Ängström exponent. We used the AERONET AOD measurements at the seven different bands (340nm, 380nm, 440nm, 500nm, 870nm and 1020nm) and apply linear regression using the above right logarithmic formula to get an estimate of the Ängström exponent $k$. The related statements were added to the revised manuscript.

Page 7, line 31. In regard to the 5 km distance between West Pasadena and Caltech, why is there not a target installed on the Caltech campus? Is the CLARS observatory to Caltech campus line of sight not possible? Line 16 on page 8 refers to other future targets. If Caltech is included in the expanded list of future targets, please mention this.

20     Thank you for this comment. Unfortunately, the Caltech campus is out of sight for the CLARS observatory near the top of Mt. Wilson, so it is not a possible future target. This statement was added to the discussion section.

Page 8, line 1. "Caltech also represents that in West Pasadena."

25     We rephrased the sentence to "we expect the AOD variations at Caltech and West Pasadena to be similar".

Page 8, line 19. The claim "We illustrate the robust ability of multi-wavelength retrievals of water vapor slant columns to provide constraints on aerosol optical properties" is not demonstrated by the current paper since optical properties are not retrieved.

30     Thank you for this comment. Many works had been done in the revised manuscript to demonstrate this claim. Figure 7 was added to show the results from implementing four controlled sensitivity tests on the correlation, between variations in $H_2O$ SCDs and AOD, relative to the two most important scattering properties, single scattering albedo and asymmetric parameter, using 2S-ESS model with input of five composite MERRA aerosols in LA basin. The results show that the variations in $H_2O$ SCDs is robustly and 35 linearly correlated with AOD, and the variations of SSA and g are not the key contributors to the variations of the standard deviations of $H_2O$ SCDs. Moreover, the rate of change of the correlation is related to the asymmetry parameter of aerosol scattering. Please refer to Section 5.2 in the revised manuscript for details.

Page 8, line 28. The proposed approach is proposed, not demonstrated, in regards to providing "a sensitive way to quantify the effect of aerosol scattering in GHG retrievals". I encourage some additional work that supports this assertion.

5    Thanks for this suggestion that led to substantial revisions in the revised manuscript. The additional work to examine the aerosol compositions in LA from MERRA reanalysis data and their corresponding aerosol scattering properties are shown in Figure 5 and Table 1, respectively, and to test the robustness of the correlation between variations in $H_2O$ SCDs and AOD and its sensitivity to aerosol scattering properties are illustrated in Figure 7.

10   To emphasize on relating this study to GHG retrievals from space, observations between 9:00 and 15:00 local time are shown, when the solar zenith angles (SZAs) are less than 60°, similar to the majority of satellite observations. This time period also includes the local overpass times at around 10:00 for SCIAMCHY (Bovensmann et al., 1999) and around 13:30 for GOSAT, OCO-2 and TanSat (Liu et al., 2011). Also, two control experiments (in Figure 6) are implemented to show the dominant contribution of
15   aerosol scattering effect to the variations in $H_2O$ SCDs.

Figure 4 and Figure 7 focus on the $H_2O$ SCDs around 13:00, the local overpass time of most current GHG observation satellites. The conclusions include, as we state in the beginning, (a) correlation between AOD and extent of variations in $H_2O$ SCDs is robust and significant, and the difference in $H_2O$ abundance and g are responsible for the difference of slopes between the winter-spring and summer-autumn periods; (b) the
20   variations of SSA and g are not the key contributors to the variations of the standard deviations of $H_2O$ SCDs; and (c) g is the main contributor to the ratio of the changes (the slope from linear regression).

30

35   **The following documents are the revised manuscript (with changes highlighted) and the supplementary figure:**

[revised manuscript text omitted]
) and the red line is linear regression result from the first 6 of the 7 bands (that is, 1020nm is excluded). Therefore, the closeness of the AOD at 1020nm to the fitted red line indicates the strength of the wavelength dependence that follows the exponential law. The averaged bias between the AOD and the red line value at 1020nm for all the 68 days is 9.82±10.02% (mean and one standard deviation).

---

## Referee Report (RR1)

Review of "Aerosol Scattering Effects on Water Vapor Retrievals over the Los Angeles Basin" by Zeng et al.

This paper uses retrievals of $H_2O$ SCDs from CLARS observations to demonstrate that standard deviations of retrieved $H_2O$ SCDs amongst 15 $H_2O$ bands are due to the effects of scattering aerosol. This fact is established by the calculations.

The results in Figure 3 are very reasonable. The panel 3a (SVO observing path from Mt. Wilson to the sun) $H_2O$ SCDs have little wavelength dependence since little aerosol impacts these observations. The panel 3b (LABS LA basin observing path) $H_2O$ SCDs display much wavelength dependence, since aerosol is not included in the retrieval, and aerosol is an important contributor to the total optical depths. The retrieval of the LABS data therefore needs to add additional $H_2O$ in lieu of aerosol that is present in the LA basin. Panel 3d indicates that the AERONET-Caltech aerosol increases during the day, and therefore the $H_2O$ SCD standard deviations in panel 3c also increase during the day.

This paper, however, is very problematic since the methodology does not go far enough in its analyses. Yes, the data in Figure 3 is consistent with the fact that aerosol impacts the total optical depths in the $H_2O$ bands. This is expected, and by itself is not a sufficient reason for publication. The current paper does not demonstrate that accurate optical properties (e.g. AODs) can be readily retrieved from the observations.

The paper should be published after additional calculations are carried out by the authors.

Major comments

The following suggested calculations would bring the paper to a level of completion that fully warrants publication:

(a) The retrieval program retrieves $H_2O$ SCDs and aerosol properties (e.g. vertical AOD) simultaneously as a function of wavelength. Representative AERONET-MERRA-GOCART SSA and g could be specified (fixed) in a daily basis in the forward model of the retrieval.

(b) Compare the retrieved aerosol properties (e.g. AOD) to those derived from a combined analysis of AERONET, MERRA, and GOCART data,

(c) The retrieval program retrieves $H_2O$ SCDs with specified AERONET-MERRA-GOCART wavelength dependent AOD, SSA, and g

 (d) Demonstrate that RT forward model calculations including the retrieved $H_2O$ and aerosol properties (from (a)) reduce the scatter in a Figure 3b type graph, and

 (e) Demonstrate that RT forward model calculations including the retrieved $H_2O$ (from (a)) and AERONET-MERRA-GOCART aerosol properties reduce the scatter in a Figure 3b type graph.

The sentence in the Abstract "The understanding of aerosol scattering effects on $H_2O$ retrievals provides a sensitive way to quantify the effect of aerosol scattering on greenhouse gas retrievals …" indicates that this fact is established by the work in this paper. The phrase "provides a

sensitive way" is not demonstrated by the work in this paper. This sentence needs to be removed from the abstract. The language on page 2, lines 19-20 is appropriate and can be retained. The language on Page 4, line 29 "shows the potential" is appropriate. The final sentence (page 10, lines 23-26) with the phrases "evidence justify our approach" and "providing a sensitive way" is not demonstrated by the paper's calculations.

Page 3, line 12. Is wavelength dependent surface reflection included in the CLARS-FTS GFIT retrieval algorithm?

Page 3, line 12. Why is wavelength dependent aerosol AOD (and possibly mean SSA and g) not included in the retrieval?

Page 3, line 21. How is Figure 2 constructed? The Figure caption refers to "normalized radiance", yet it is not stated in the paper how the normalized radiance is calculated. Please do so. Is aerosol included in the RT model calculations that are presented in Figure 2?

Page 3, lines 24-25. A full information analysis would follow the Rodgers methodology and include (a) calculations using a retrieval state vector that includes both $H_2O$ and aerosol (with little influence by the *a priori* aerosol) and (b) calculations using a state vector that includes $H_2O$ and aerosol heavily constrained by the AERONET-MERRA-GOCART data.

Page 6 and Figure 6. It is requested that panel (a) also be presented with the means included. The results in panels (a) and (c) are opposite to what is expected. If one adds the AERONET data to the forward model RT calculation, I would expect that the scaling factors would be closer to unity (and closer to 0 when the means are subtracted from the scaling factors) than for the case when no aerosol (panel (c)) is included in the forward model RT calculations. Yet the opposite is apparent. Please clarify.

Page 6 and Figure 6. If the standard deviations are not normalized, what does panel 6d look like? Again, the Aerosol-Free and Clear day curves seem to indicate that less *a priori* information (and/or a less complete inclusion of all contributors to the forward model) produces a better result. Please clarify.

Minor comments

Page 3, line 27. It is important to mention that aerosol is also included in the forward model RT.

Page 4, line 1 Briefy mention how IC is calculated.

Page 4, line 3. State which variables are retrieved.

Page 4, lines 17-18. Indicate (in %) the representative "small differences" and "larger variation" values.

Page 5, line 11. State the wavelength range "wavelengths (i.e. from 1288 nm to 2190 mn))"

Page 5, line 12. The sentence is not clear. Is the "AOD data" a vertical optical depth, and the AOP is simply this value scaled by a SZA and CLARS viewing angle geometric airmass factor?

Page 5, line 21. Why should the PBLH be similar throughout the year? Is the PBLH information available from nearby airport radiosonde temperature-pressure profiles or from any camera images taken at CLARS? On page 6 the PBLH data is from spring 2010, yet the observations are in winter-spring and summer-autumn 2013. Explain why (and if) the PBLH values are so uniform. On page 7 (line 1) it is stated that the PBLH is an important parameter. Should it not be included in the retrieval state vector?

Page 5, line 23. The "other factors" are not discussed in the text. What are the "other factors"?

Page 6, line 13. Why is HITRAN 2008 (instead of HITRAN 2012) used in the RT calculations?

Page 6, line 14. Is the surface albedo of 0.23 really wavelength independent from 1288 nm to 2190 nm?

Page 8, line 27. Is annual average MERRA data or month-specific MERRA data used for each individual day of the 68 day set?

---

## Referee Report (RR2)

Review of "Aerosol Scattering Effects on Water Vapor Retrievals over the Los Angeles Basin" by Zeng et al.

The authors have made a strong effort to add to the paper in response to the suggestions of my 2nd review. Figure 8 of the revised paper addresses the major portion of the previous suggestions. The new lines added to the text, e.g. lines 1-8 on page 11, nicely bring a sense of closure to the paper. The paper can now be published subject to minor revisions.

Minor comments

Page 3, lines 5-7 This sentence needs to be revised i.e. "The CLARS measurement technique from Mt. Wilson mimics geostationary satellite observations of reflected sunlight, which are governed by a sun to surface and surface to instrument optical path geometry. The geostationary observations can be used to retrieve GHG mixing ratios (Xi et al., 2015)."

Page 3, line 23. "efficiency of band channels" is an odd phrase to use. Perhaps revise to "is a powerful tool which can be used in the channel selection process"

Page 5, lines 17-19. The sentence needs to be rewritten i.e. "Furthermore, aerosol optical path (AOP) values are calculated by multiplying the vertical path AOD data by air mass factors, which are derived from the SZA and viewing zenith angle of CLARS at West Pasadena (83.1°)."

Pages 8 and 9. When I read these pages I naturally searched for a way to make sense of the writing. The current text has run-on paragraphs. The pages should be broken up into coherent paragraphs. Once I did this, the writing was easier to assimilate. Suggested paragraph breaks:

Page 8, line 6 "We perform three experiments.."

Page 8, line 14. "The results for simulated $H_2O$.."

Page 8, line 24. "Further confirmation of this …"

Page 9, line 14. "The result is shown in.."

Page 9, line 22. "Similar control experiments.."

Page 10, line 12. Delete "for the purpose of providing comparable results for future satellite study" or revise. The phrase "comparable results" is not clear.

---

## Author Response (AR2)

Item-by-item responses to the specific comments are provided below, in which the reviewers' comments are in **blue**, our responses in **black**, and modifications to the original manuscript are indicated by highlight in yellow. The revised manuscript with highlighted changes is attached at the end.

Co-Editor Decision: Reconsider after major revisions (22 Nov 2016) by AE Anne Perring

Comments to the Author:

Dear authors,

Both reviewers still have significant concerns about the findings in this paper and recommend further revisions. Reviewer #2, especially, recommends some specific items which would make the work more convincing. Please consider these comments carefully during your revisions. Once these concerns have been addressed I will be happy to reconsider publication.

Anne

Dear Editor,

We thank the reviewers' thoughtful comments and constructive suggestions. We have made careful, major revisions following the comments and suggestions from the reviewers. Please refer to the item-by-item responses for details.

Regards,

The Authors.

Item-by-item responses to the specific comments are provided below, in which the reviewers' comments are in blue, our responses in **black**, and modifications to the original manuscript are indicated by highlight in yellow. The revised manuscript with highlighted changes is attached at the end.

Reviewer #1

Minor comments/suggestions:

I am still not convinced of how the authors claim that there is a strong wavelength dependency in the aerosol they examine (as shown in the auxiliary data), but then show that the majority of aerosols are dominated by sea salt and dust, which are relatively large particles with small wavelength dependency (but with indeed large asymmetry parameter). Maybe it would be clearer to state that the AOD's were extrapolated to the FTS wavelengths using the Angstrom exponent assumption, for the sake of the RT calculations. Suggesting large wavelength dependency for such aerosol types is a bit unclear. What are the extrapolated AOD levels along the FTS wavelengths (I am assuming they show very weak wavelength dependency, which is ok but just has to be stated)?

Thank you for the comment. The AOD data from AERONET is the total AOD of different aerosol types, which makes it difficult to separately check wavelength dependency for different types. However, from the AERONET AOD measurements, as shown in **Supplemental Material Figure 1** (the calculated Ångström coefficients were added), we can see the wavelength dependency in the AERONET-AOD measurement bands from 340 to 1020 nm follows the Ångström exponent law. The interpolated AOD levels along the $H_2O$ absorption bands for all the 68 days can be seen in Figure 8(a). For measurements at the same time, the interpolated AOD levels at 7760 cm$^{-1}$ $H_2O$ bands are close to the AERONET AOD at 1020nm, but at the 4554 cm$^{-1}$ $H_2O$ band it is about 2-3 times less. For all the interpolated AOD in the 15 $H_2O$ bands, even though the dependency becomes weaker in the $H_2O$ bands than that in the AERONET bands, there is still difference of AOD between different bands.

Following your suggestions, in the revised manuscript, we added/rephrased the following statements in Section 5:

"For the sake of calculations in the 2S-ESS RT model, the AOD data in these 15 bands were extrapolated using the Ångström exponent law (Seinfeld and Pandis, 2006; Zhang et al., 2015)."

And,

"Examples of applying this law to the AERONET AOD measurements are shown in Supplemental Material Figure 1, from which we can see that the wavelength dependence of total AOD, a combination of different types of aerosols, generally follows the above exponent law."

Please add colorbar title (i.e. Month) to Fig. 7

Done. Thanks for pointing this out.

Item-by-item responses to the specific comments are provided below, in which the reviewers' comments are in **blue**, our responses in **black**, and modifications to the original manuscript are indicated by highlight in yellow. The revised manuscript with highlighted changes is attached at the end.

**Reviewer #2**

Review of "Aerosol Scattering Effects on Water Vapor Retrievals over the Los Angeles Basin" by Zeng et al.

This paper uses retrievals of H2O SCDs from CLARS observations to demonstrate that standard deviations of retrieved H2O SCDs amongst 15 H2O bands are due to the effects of scattering aerosol. This fact is established by the calculations.

The results in Figure 3 are very reasonable. The panel 3a (SVO observing path from Mt. Wilson to the sun) H2O SCDs have little wavelength dependence since little aerosol impacts these observations. The panel 3b (LABS LA basin observing path) H2O SCDs display much wavelength dependence, since aerosol is not included in the retrieval, and aerosol is an important contributor to the total optical depths. The retrieval of the LABS data therefore needs to add additional H2O in lieu of aerosol that is present in the LA basin. Panel 3d indicates that the AERONET-Caltech aerosol increases during the day, and therefore the H2O SCD standard deviations in panel 3c also increase during the day.

This paper, however, is very problematic since the methodology does not go far enough in its analyses. Yes, the data in Figure 3 is consistent with the fact that aerosol impacts the total optical depths in the H2O bands. This is expected, and by itself is not a sufficient reason for publication. The current paper does not demonstrate that accurate optical properties (e.g. AODs) can be readily retrieved from the observations.

The paper should be published after additional calculations are carried out by the authors.

Thank you for your suggestions. We have made major revisions following your comments and suggestions; the main changes in the revised manuscript include the following:

(1) We performed the suggested calculation to retrieve $H_2O$ and AOD simultaneously using the $H_2O$ bands in a realistic numerical simulation study to demonstrate that accurate AODs can be retrieved from the spectral data, and to investigate the impact of AOD retrieval on the process of retrieving $H_2O$ SCDs.

(2) Four figures were added, including **Supplemental Material Figure 2** to show the retrieval work flow, **Supplemental Material Figures 3 and 4** to provide more details about the dominant aerosol scattering effect on $H_2O$ SCD retrievals (in Figure 6), and **Figure 8** to show the results from your suggested calculations.

Please see below for our item-by-item responses to your suggestions and comments.

**Major comments:**

The following suggested calculations would bring the paper to a level of completion that fully warrants publication:

(a) The retrieval program retrieves H2O SCDs and aerosol properties (e.g. vertical AOD) simultaneously as a function of wavelength. Representative AERONET-MERRA-GOCART SSA and g could be specified (fixed) in a daily basis in the forward model of the retrieval.

(b) Compare the retrieved aerosol properties (e.g. AOD) to those derived from a combined analysis of AERONET, MERRA, and GOCART data,

(c) The retrieval program retrieves H2O SCDs with specified AERONET-MERRA-GOCART wavelength dependent AOD, SSA, and g

(d) Demonstrate that RT forward model calculations including the retrieved H2O and aerosol properties (from (a)) reduce the scatter in a Figure 3b type graph, and

(e) Demonstrate that RT forward model calculations including the retrieved H2O (from (a)) and AERONET-MERRA-GOCART aerosol properties reduce the scatter in a Figure 3b type graph.

We thank the reviewer for this comment, which led us to make substantial improvements to the manuscript. Given the fact that the operational retrieval algorithm for CLARS-FTS developed based on the GFIT algorithm does not take aerosol scattering into account, here we make the calculations suggested by the reviewer by conducting a realistic numerical simulation study using the 2S-ESS RT model. The advantages of this numerical simulation study are that (1) the truth state vector is known and we can directly assess the accuracy of the retrievals, and (2) control experiments can be conducted by minimizing the influences from other factors except the one that is investigated. The purposes of these calculations are to (1) demonstrate that accurate AODs can be retrieved from the spectral data, and (2) investigate the impact of AOD retrieval on the process of retrieving $H_2O$ SCDs. The calculations are implemented using the noon data for the 68 days shown in Figure 4, for the purpose of providing comparable results for future satellite study. To describe the retrieval procedure described in Section 5, we show in **Supplementary Material Figure 2** the schematic diagram of the retrieval algorithm which is based on the 2S-ESS RT model and Bayesian inversion theory.

The calculations and results (shown in **Figure 8**) are as follows:

(1) Simultaneous retrieval of $H_2O$ SCDs and AODs

In the retrieval algorithm as shown in **Supplementary Material Figure 2**, we first produce synthetic spectra using the 2S-ESS RT model with input AOD from AERONET and SSA and g from MERRA-GOCART. We then set $H_2O$ SCD scale factor and AOD as the state variables, and simultaneously retrieve them using the 15 $H_2O$ absorption bands. The results are shown in Figure 8(a) and 8(b). From the comparison between retrieved and true AODs (Figure 8(a)), we see that the retrieved AOD agrees well with the truth in the $H_2O$ absorption bands for all months ($R^2$=0.93; RMSE=0.0051). From the simultaneously retrieved $H_2O$ SCD scale factor and AOD averaged over the 15 bands (Figure 8(b)), we see that their retrieval errors show a similar pattern; when the $H_2O$ SCD scale factor diverges from unity, the difference between the retrieved and true AOD also increases. When $H_2O$ SCD becomes smaller, the observed radiance increases, since fewer photons are absorbed in the $H_2O$ bands. On the other hand, when AOD becomes smaller, the observed radiance decreases since the aerosol scattering effect becomes weaker and fewer photons will be scattered to the observer in the $H_2O$ bands. Therefore, we can see that the AOD and

H$_2$O SCD retrievals show the same pattern, whereby smaller AOD retrievals coincide with smaller H$_2$O SCD retrievals, so that their effects on the observed radiance largely cancel out.

(2) Retrieval of H$_2$O SCDs when AODs are perfectly known

We set the H$_2$O SCD to be the only state variable and assume that the AOD is perfectly known. Comparison of the retrieved H$_2$O SCD scale factors from three different cases are shown in Figure 8(c). In Case A, aerosol scattering is not considered; in Case B, H$_2$O and AOD are simultaneously retrieved; in Case C, the AOD is perfectly known. We can see that (1) H$_2$O SCD can be accurately retrieved if we have perfect knowledge of AOD, and (2) when we retrieved AOD and H$_2$O SCD simultaneously, the variations in retrieved H$_2$O SCD scale factors are largely reduced compared to the case when aerosol scattering is not considered.

Based on the above results from a realistic numerical simulation study, we conclude that (1) accurate AODs and H$_2$O SCDs can be simultaneously retrieved from the spectral data in the 15 H$_2$O absorption bands, and (2) variations in retrieved H$_2$O SCD scale factors are largely reduced when we retrieve AOD simultaneously compared to that when aerosol scattering is not considered.

The above statements are added to the revised manuscript (Section 5.3).

[Figure]

**Supplementary Material Figure 2.** Schematic diagram of the retrieval algorithm based on the 2S-ESS RT model and Bayesian inversion theory. A detailed description is provided in Section 5.

[Figure]

**Figure 8**. Retrieval of AOD and $H_2O$ SCD simultaneously based on a realistic numerical simulation study using the 2S-ESS RT model. (a) Scatter plot between true and retrieved AOD from the simultaneous retrieval experiment. The mean and standard deviation of the difference between them are −0.0018 and 0.0051, respectively. The black dotted line is the one-to-one line; (b) Retrieval of AOD and $H_2O$ SCD scale factors averaged over the 15 $H_2O$ absorption bands from the simultaneous retrieval experiment; the one sigma error bar is also shown; (c) $H_2O$ SCD scale factor retrievals from three different cases; in Case A, aerosol scattering is not considered; in Case B, $H_2O$ and AOD are simultaneously retrieved; in Case C, AOD is perfectly known.

The sentence in the Abstract "The understanding of aerosol scattering effects on H2O retrievals provides a sensitive way to quantify the effect of aerosol scattering on greenhouse gas retrievals …" indicates that this fact is established by the work in this paper. The phrase "provides a sensitive way" is not demonstrated by the work in this paper. This sentence needs to be removed from the abstract. The language on page 2, lines 19-20 is appropriate and can be retained. The language on Page 4, line 29 "shows the potential" is appropriate. The final sentence (page 10, lines 23-26) with the phrases "evidence justify our approach" and "providing a sensitive way" is not demonstrated by the paper's calculations.

Thank you for these suggestions. "provides a sensitive way" was changed to "suggests a promising way"; Based on the results from your suggested calculations on retrieving AOD using $H_2O$ absorption bands, shown in Section 5.3, our proposed approach was shown to be effective in deriving the aerosol scattering effect. Therefore, in the revised manuscript, we keep the sentence "…evidence justify our proposed approach…".

Page 3, line 12. Is wavelength dependent surface reflection included in the CLARS-FTS GFIT retrieval algorithm?

In the CLARS-FTS GFIT retrieval algorithm, we assume the surface reflectivity to be constant across the 15 $H_2O$ absorption bands (from 4554 to 7760 $cm^{-1}$, that is, 1288 to 2195 nm) with a value of 0.23, as measured for West Pasadena (Fu et al., 2014).

The surface reflection is probably wavelength dependent; however, we make that assumption for the sake of simplicity, and since the albedo is not the focus of this study. The key assumption is not that the surface albedo is constant; it is that the spectral dependence of the albedo is reasonably well known. In the authors' opinion, this is not an unreasonable assumption.

Page 3, line 12. Why is wavelength dependent aerosol AOD (and possibly mean SSA and g) not included in the retrieval?

The version 1.0 operational retrieval algorithm of CLARS-FTS (Fu et al., 2014) was developed based on the Gas FItting Tool (GFIT) algorithm, which does not take aerosol scattering into account and is numerically efficient and particularly applicable to massive data processing when computational resources are limited (for details, please see Toon et al. (1992) and Wunch et al. (2011)). When aerosol loading is not negligible, as in the LA basin, the CLARS measurements retrieved from GFIT provides suitable data source for examining the aerosol scattering effects on gas retrievals, as our study shows using $H_2O$ SCD retrievals.

However, in our simulations using the 2S-ESS RT model to generate synthetic spectra, AOD from AERONET and SSA and g from MERRA-GOCART are incorporated in the forward modeling.

Page 3, line 21. How is Figure 2 constructed? The Figure caption refers to "normalized radiance", yet it is not stated in the paper how the normalized radiance is calculated. Please do so. Is aerosol included in the RT model calculations that are presented in Figure 2?

The "normalized radiance" is obtained by dividing the spectra by the maximum radiance value, so the radiance ranges between 0 and 1. We added the related statements in the revised manuscript.

Yes, the RT model calculations shown in Figure 2 include AOD observations from AERONET, and SSA and g from MERRA-GOCART. This statement was added in the revised manuscript.

Page 3, lines 24-25. A full information analysis would follow the Rodgers methodology and include (a) calculations using a retrieval state vector that includes both H2O and aerosol (with little influence by the a priori aerosol) and (b) calculations using a state vector that includes H2O and aerosol heavily constrained by the AERONET-MERRA-GOCART data.

Thanks for the suggestion. In Section 5.3 of the revised manuscript, we added the ICs for the 15 bands from the calculations of simultaneous retrieval of $H_2O$ SCD and AOD (Table 1). The aerosol scattering properties used in the retrieval program, including SSA and g, are obtained from MERRA and GOCART.

The new ICs for the 15 $H_2O$ bands are (from 4554 to 7760 $cm^{-1}$):

[8.61, 8.47, 8.27, 6.72, 8.04, 9.55, 8.11, 6.29, 5.36, 7.58, 8.73, 9.40, 7.61, 8.23, 7.72]

Page 6 and Figure 6. It is requested that panel (a) also be presented with the means included. The results in panels (a) and (c) are opposite to what is expected. If one adds the AERONET data to the forward model RT calculation, I would expect that the scaling factors would be closer to unity (and closer to 0 when the means are subtracted from the scaling factors) than for the case when no aerosol (panel (c)) is included in the forward model RT calculations. Yet the opposite is apparent. Please clarify.

**Supplementary Material Figure 3** shows the same result as Figure 6(a), except that the mean is not subtracted. We can see that, as is expected and consistent with CLARS measurements, the variation in retrieved $H_2O$ SCD scaling factors from multiple bands increases from the morning to the afternoon, consistent with the increasing trend of the aerosol loading.

We apologize for the misunderstanding. The interaction between forward and inverse modeling can be more clearly seen in Supplementary Material Figure 2**.** The work flow includes: (1) using the 2S-ESS RT model to generate synthetic spectral radiance data for the 15 chosen bands with aerosol properties from AERONET-MERRA-GOCART; (2a) for retrieving $H_2O$ SCD, fitting the synthetic spectral data based on Bayesian inversion theory (Rodgers, 2000) using the forward 2S-ESS RT model with the same configuration, but with AOD set to zero and held constant, as in Zhang et al. (2015). This approach approximately simulates the influence of neglecting aerosol scattering on the retrieved $H_2O$ SCDs by CLARS; or (2b) for simultaneously retrieving $H_2O$ SCD and AOD, fitting the synthetic spectral data based on Bayesian inversion theory using the forward 2S-ESS RT model with the same configuration, but AOD set as a state variable along with $H_2O$ SCD.

In Figure 6, the change of AOD in the forward modeling is made only when we generate the synthetic spectral radiance; for inverse modeling (when we use the RT model to produce a Jacobian matrix), we keep the AOD constant at zero to approximately simulate the influence of neglecting aerosol scattering on the retrieved $H_2O$ SCDs by CLARS. Therefore, in Figure 6(a), when we add the AERONET data to the forward model RT calculation to generate the synthetic spectra, but neglect aerosol scattering in the retrieval algorithm, we can see the impact of neglecting aerosol scattering on the $H_2O$ retrievals. This is the reason we see large variations in $H_2O$ SCD. In Figure 6(c), we do not include aerosol in the forward modeling when generating synthetic spectra; therefore, when we use a retrieval algorithm that neglects aerosol scattering, we can accurately retrieve the $H_2O$ SCDs. This is the reason we see scaling factors close to unity in Figure 6(c). We added the above statements to the revised manuscript.

[Figure]

**Supplementary Material Figure 3**. Scaling factors for $H_2O$ SCDs retrieved from the simulated synthetic spectral radiances in the 15 chosen bands using the 2S-ESS RT model with AOD data from AERONET-Caltech on March 01, 2013. This plot is the same as Figure 6(a), except that the mean is not subtracted.

Page 6 and Figure 6. If the standard deviations are not normalized, what does panel 6d look like? Again, the Aerosol-Free and Clear day curves seem to indicate that less a priori information (and/or a less complete inclusion of all contributors to the forward model) produces a better result. Please clarify.

The plots are shown in **Supplementary Material Figure 4**, where (a) is the standard deviation of $H_2O$ SCD retrievals from CLARS, and (b) is the standard deviation of $H_2O$ SCD scaling factor simulated by the 2S-ESS RT model (for the same three cases as in Figure 6). We can see that the data show the same pattern.

As in our response to your previous comment, the AOD data for aerosol-free and clear-day scenarios are only used in the forward model RT calculation to generate the synthetic spectra. But when implementing the retrieval algorithm, we do not consider the aerosol scattering (set AOD=0) in order to mimic the CLARS retrieval algorithm and examine the impact of neglecting aerosol scattering on the $H_2O$ SCD retrievals. So for the aerosol-free case, we can retrieve the state variation of $H_2O$ SCD with better precision since the generated spectra is not contaminated by aerosol scattering, than that for the clear-day case, in which the generated spectra is still influenced by weak aerosol scattering.

[Figure]

**Supplementary Material Figure 4**. Standard deviation of $H_2O$ (a) SCDs observed by CLARS (unit: molecules), and (b) SCD scaling factors simulated by the 2S-ESS RT model. The three cases in (b) are the same as those in Figure (6).

**Minor comments**

Page 3, line 27. It is important to mention that aerosol is also included in the forward model RT.

Thanks for this comment. The sentence "AOD observations from AERONET, and SSA and g from MERRA-GOCART are included in the RT model" was added in the revised manuscript.

Page 4, line 1 Briefly mention how IC is calculated.

The Information Content (H) can be calculated by the following equation adopted from Rodgers (2000) (equations (2.79) and (2.80)):

$$H = -\frac{1}{2}\ln|\mathbf{I} - \mathbf{A}| = -\frac{1}{2}\ln|(\mathbf{K}^T\mathbf{S}_\epsilon^{-1}\mathbf{K} + \mathbf{S}_a^{-1})^{-1}\mathbf{S}_a^{-1}|$$

where **A** is the averaging kernel, a measurement of sensitivity of the retrieval state to true state variables; **I** is the identity matrix; **K** is the Jacobian, a weighting function matrix which measures the sensitivity of measured radiance to state variables of gas abundances; and $\mathbf{S}_a$ and $\mathbf{S}_\epsilon$ are the error covariance matrixes for *a priori* and measurement data respectively.

We added the following statements in the revised manuscript: "IC is calculated as $-\ln|\mathbf{I} - \mathbf{A}|/2$, where **I** is the identity matrix and **A** is the averaging kernel matrix, a measure of the sensitivity of the retrieval state to true state variables.".

Page 4, line 3. State which variables are retrieved.

Thank you for the comment. The retrieved trace gas variables include $CO_2$, $CH_4$, $CO$, and $N_2O$.

Page 4, lines 17-18. Indicate (in %) the representative "small differences" and "larger variation" values.

Thank you for the comment. In the revised manuscript, to quantify the "small differences", we added "about 14.2% and 8.2% of those from West Pasadena for March 01 and September 28, respectively, in terms of standard deviation of $H_2O$ SCD retrievals"; And to quantify the "larger variation", we added "about 7 times and 12 times those from SVO for March 01 and September 28, respectively, in terms of standard deviation of $H_2O$ SCD retrievals".

Page 5, line 11. State the wavelength range "wavelengths (i.e. from 1288 nm to 2190 mn))"

Thanks for the comment. It was added in the revised manuscript.

Page 5, line 12. The sentence is not clear. Is the "AOD data" a vertical optical depth, and the AOP is simply this value scaled by a SZA and CLARS viewing angle geometric airmass factor?

Thanks for the comment. Yes, AOD data are vertical measurements and AOP is the scaled value by an air mass factor calculated from solar and CLARS geometries. In the revised manuscript, we rephrased the sentence to "the AOD data (which are vertical measurements) are scaled to the optical depths along the slant path, to produce the aerosol optical path (AOP), by an air mass factor derived from the SZA and viewing zenith angle of CLARS at West Pasadena (83.1°)."

Page 5, line 21. Why should the PBLH be similar throughout the year? Is the PBLH information available from nearby airport radiosonde temperature-pressure profiles or from any camera images taken at CLARS? On page 6 the PBLH data is from spring 2010, yet the observations are in winter-spring and summerautumn 2013. Explain why (and if) the PBLH values are so uniform. On page 7 (line 1) it is stated that the PBLH is an important parameter. Should it not be included in the retrieval state vector?

Thanks for the question. The following Figure i shows the averaged daily PBLH measurements collected during the intensive CalNex-LA ground campaign from May 15 to June 15 in 2010 (Newman et al., 2013). We can see that during the late morning and early afternoon time from 11 am to 3 pm, the PBLH is relatively stable with small variation, even though the temperature changes during this time period. This might indicate that the PBLH at noon during the two time periods (winter-spring and summer-autumn), in general, does not have a large difference.

The PBLH is not measured by CLARS, and the measurements are not available from other data sources, like airport radiosonde profiles as you mentioned. The PBLH is indeed important for modeling aerosol scattering effects; however, the current CLARS measurements (SNR=300; spectral resolution=0.06cm$^{-1}$) may not sensitive enough to accurately retrieve PBLH.

[Figure]

Figure i. The averaged daily PBLH measurements collected during the intensive CalNex-LA ground campaign from May 15 to June 15 in 2010 (Newman et al., 2013). The plot was adopted from Figure 3(b) of Newman et al. (2013).

Page 5, line 23. The "other factors" are not discussed in the text. What are the "other factors"?

Thanks for the question. "Other factors" include factors that change during the day which may have effects on the variations of $H_2O$ SCD retrievals, including changes in solar geometry and PBLH. In the revised manuscript, we rephrased the sentence by adding "…, such as solar geometry and PBLH, …".

Page 6, line 13. Why is HITRAN 2008 (instead of HITRAN 2012) used in the RT calculations?

Thanks for the question. We used HITRAN 2008 in 2S-ESS to be consistent with the GFIT retrieval program used by CLARS, which is based on HITRAN 2008.

Page 6, line 14. Is the surface albedo of 0.23 really wavelength independent from 1288 nm to 2190 nm?

It is probably not; however, we make that assumption for the sake of simplicity, and since the albedo is not the focus of this study. The key assumption is not that the surface albedo is constant; it is that the spectral

dependence of the albedo is reasonably well known. In the authors' opinion, this is not an unreasonable assumption.

Page 8, line 27. Is annual average MERRA data or month-specific MERRA data used for each individual day of the 68 day set?

Thanks for the question. The MERRA AOD reanalysis are available at three times each day: 10:00, 13:00 and 16:00 (Connor et al., 2016). For each of the 68 days, we use the daily MERRA data at 13:00. We added the following sentence in Section 5.2: "…daily aerosol compositions (at 13:00 local time; monthly mean 
[revised manuscript text omitted]

---

## Author Response (AR3)

Item-by-item responses to the specific comments are provided below, in which the reviewers' comments are in **blue**, our responses in **black**, and modifications to the original manuscript are indicated by highlight in yellow. The revised manuscript with highlighted changes is attached at the end.

Co-Editor Decision: Reconsider after minor revisions (Editor review) (26 Jan 2017) by AE Anne Perring
Comments to the Author:
Dear authors,
Please see a few remaining minor comments from Referee #2. Once these have been dealt with I will be happy to recommend your manuscript for publications.
Anne

Dear Editor,

We thank the reviewer's thoughtful comments. We have made careful, minor revisions following the comments from the reviewer. Please refer to the item-by-item responses for details.

Regards,
The Authors

Review of "Aerosol Scattering Effects on Water Vapor Retrievals over the Los Angeles Basin" by Zeng et al.

The authors have made a strong effort to add to the paper in response to the suggestions of my 2nd review. Figure 8 of the revised paper addresses the major portion of the previous suggestions. The new lines added to the text, e.g. lines 1-8 on page 11, nicely bring a sense of closure to the paper. The paper can now be published subject to minor revisions.

Minor comments

Page 3, lines 5-7 This sentence needs to be revised i.e. "The CLARS measurement technique from Mt. Wilson mimics geostationary satellite observations of reflected sunlight, which are governed by a sun to surface and surface to instrument optical path geometry. The geostationary observations can be used to retrieve GHG mixing ratios (Xi et al., 2015)."
Thanks for the comment. It was changed in the revised manuscript.

Page 3, line 23. "efficiency of band channels" is an odd phrase to use. Perhaps revise to "is a powerful tool which can be used in the channel selection process"
Thanks. Done.

Page 5, lines 17-19. The sentence needs to be rewritten i.e. "Furthermore, aerosol optical path (AOP) values are calculated by multiplying the vertical path AOD data by air mass factors, which are derived from the SZA and viewing zenith angle of CLARS at West Pasadena (83.1°)."
Thanks for the suggestion. The sentence had been revised according to your suggestion.

Pages 8 and 9. When I read these pages I naturally searched for a way to make sense of the writing. The current text has run-on paragraphs. The pages should be broken up into coherent paragraphs. Once I did this, the writing was easier to assimilate. Suggested paragraph breaks:
Page 8, line 6 "We perform three experiments.."
Page 8, line 14. "The results for simulated H2O.."
Page 8, line 24. "Further confirmation of this …"
Page 9, line 14. "The result is shown in.."
Page 9, line 22. "Similar control experiments.."
Thanks for this thoughtful suggestion. We broke up the paragraphs as you suggested.

Page 10, line 12. Delete "for the purpose of providing comparable results for future satellite study" or revise. The phrase "comparable results" is not clear.
We deleted the sentence to avoid any confusion it may cause. Thanks.

[revised manuscript text omitted]